

# Impacts of the Denver Cyclone on Regional Air Quality and Aerosol Formation in the Colorado Front Range during FRAPPÉ 2014

Kennedy T. Vu[1], Justin H. Dingle[1], Roya Bahreini[1,2], Patrick J. Reddy[3†], Teresa L. Campos[3], Glenn S. Diskin[4], Alan Fried[5], Scott C. Herndon[6], Rebecca S. Hornbrook[3], Greg Huey[7], Lisa Kaser[3], Denise D. Montzka[3], John B. Nowak[6], Dirk Richter[5], Joseph R. Roscioli[6], Stephen Shertz[3], Meghan Stell[3], David Tanner[7], Geoff Tyndall[3], James Walega[5], Peter Weibring[5], Andrew J. Weinheimer[3], Gabriele Pfister[3], Frank Flocke[3]

[1] Environmental Toxicology Graduate Program, University of California, Riverside, CA 92521
[2] Department of Environmental Sciences, University of California, Riverside, CA 92521
[3] Atmospheric Chemistry Observations & Modeling Laboratory, National Center for Atmospheric Research, Boulder, CO 80301
[4] Chemistry and Dynamics Branch, NASA Langley Research Center, Hampton, VA 23681
[5] Institute for Arctic and Alpine Research, University of Colorado, Boulder, CO 80303
[6] Aerodyne Research, Inc., Billerica, MA 01821
[7] Department of Earth and Atmospheric Sciences, Georgia Institute of Technology, Atlanta, GA 30033
[†] Visitor at NCAR, Boulder, CO 80301

*Correspondence to*: R. Bahreini (Roya.Bahreini@ucr.edu)

**Abstract.** We present airborne measurements made in the Colorado Front Range aboard the NSF C-130 aircraft during the 2014 Front Range Air Pollution and Photochemistry Éxperiment (FRAPPÉ) project. Data on trace gases, non-refractory sub-micron aerosol chemical constituents, and aerosol optical extinction ($\beta_{ext}$) at λ=632 nm in the presence and absence of a surface mesoscale circulation pattern, called the Denver Cyclone, were analyzed in three study regions of the Front Range: In-Flow, Northern Front Range (NFR), and Denver Metropolitan (DM). Pronounced increases in mass concentrations of organics, nitrate, and sulfate in NFR and DM were observed during the cyclone episodes (27-28 July) compared to the non-cyclonic days (26 July, 02-03 August). Organics (OA) dominated the mass concentrations on all evaluated days, with a 45% increase in OA on cyclone days across all three regions while the increase during the cyclone episode was up to ~80% for DM, from 3.78±1.55 µg sm$^{-3}$ to 6.78±1.78 µg sm$^{-3}$, where sm$^{-3}$ is the STP unit of volume of air. Average nitrate mass concentrations were 0.26 ± 0.27 µg sm$^{-3}$ vs. 1.03±0.74 µg sm$^{-3}$ followed by sulfate with an average of 0.58±0.23 µg sm$^{-3}$ vs. 1.08±0.73 µg sm$^{-3}$ on non-cyclone vs. cyclonic days, respectively. In the most aged air masses (NO$_x$/NO$_y$<0.5), background OA over DM increased by a factor of ~4, from 0.93±0.33 µg sm$^{-3}$ to 3.70±0.28 µg sm$^{-3}$ due to transport from NFR. Furthermore, enhanced partitioning of nitric acid to the aerosol phase was observed during the cyclone episodes, mainly due to increased abundance of gas phase NH$_3$. During the non-cyclone events, $\beta_{ext}$ displayed strong correlations (r=0.71) with OA and NO$_3^-$ in NFR and only with OA (r=0.70) in DM while correlation of $\beta_{ext}$ during the cyclone was strongest (r=0.86) with NO$_3^-$ in DM. Mass extinction efficiency values (MEE) values in DM were similar under cyclone (2.85±0.63 m$^2$ g$^{-1}$) and



non-cyclone ($2.72\pm0.61$ m$^2$ g$^{-1}$) days despite the dominant influence of different aerosol species on $\beta_{ext}$ (non-cyclone: OA, cyclone; NO$_3^-$).

## 1 Introduction

Atmospheric aerosols are of interest due to their impacts on human health, visibility, and climate radiative forcing through scattering and absorption of solar radiation (Monks *et al.*, 2009; Stocker *et al.*, 2013). Notably, numerous studies have shown that aerosols contribute to respiratory and cardiac disease, leading to an increase in morbidity and mortality in humans (Dockery *et al.*, 1993; Dockery and Schwartz, 1995; Pope *et al.*, 1995; Pope III *et al.*, 1995; Bascom *et al.*, 1996; Pope *et al.*, 2002; Poschl, 2005; Valavanidis *et al.*, 2008; Pope *et al.*, 2009). Moreover, ecological changes in lakes and national forests from nitrogen deposition are a driving concern for the sustainability of the ecosystem (Wilson and Spengler, 1996; Baron *et al.*, 2000; Williams and Tonnessen, 2000; Blett *et al.*, 2004; Burns, 2004; Seinfeld and Pandis, 2012).

Urban air is comprised of a highly complex mixture of gaseous and particulate pollutants, including volatile organic compounds (VOCs), nitrogen oxides (NO and NO$_2$), sulfur dioxide (SO$_2$), ozone (O$_3$) and fine particulate matter (PM$_{2.5}$), and is detrimental to the environment and well-being of the public. A significant amount of submicron aerosol mass in the troposphere is comprised of organic aerosols (OA), but direct sources, composition, and formation processes of OA are still not fully understood (Pandis *et al.*, 1992; Turpin and Huntzicker, 1995; Odum *et al.*, 1996; Schell *et al.*, 2001; Claeys *et al.*, 2004; Kroll *et al.*, 2006; Volkamer *et al.*, 2006; Kroll and Seinfeld, 2008; Hallquist *et al.*, 2009; Jimenez *et al.*, 2009; Zhang *et al.*, 2011). Generally, OA are comprised of primary emitted particles into the atmosphere (i.e., primary organic aerosols; POA) and products formed from multiphase chemical reactions as secondary organic aerosols (SOA). Several important factors including aerosol composition and size determine the extent to which aerosols affect the environment and health.

The Colorado Front Range continues to face challenges attributed to air quality. In 2007, the Northern Front Range (NFR) and the Denver Metropolitan area (DM) were designated as federal non-attainment areas for the federal 8-h ozone standard (75 ppbv), averaged over three years (EPA, 2008). Since May 2016, this area is classified as a "moderate" nonattainment regions for failure to attain the federal 8-h ozone standard of 75 ppbv (averaged over three years) (Fed.Register, 2016). Furthermore, under the Clean Air Act, the U.S. EPA Regional Haze Rule mandates the reduction in anthropogenic emissions to achieve visibility improvement in wilderness areas, including Colorado's Rocky Mountain National Park. Additionally, the State of Colorado has implemented a visibility standard based on optical extinction of 76 Mm$^{-1}$, averaged within a 4-h period when relative humidity (RH) is less than 70%. This measure of total optical extinction is provided by an Optec LRT-2 long–range transmissometer at 550 nm between east Denver and downtown (39°44'8.52"N, 104°57'29.50"W) from 8:00-16:00 (MST in winter and MDT in summer). The establishment of the Denver visibility standard-setting is covered in detail by Ely *et al.* (1993).

Vehicular emissions from growing urbanization in the Denver Metropolitan area, local power-plants, agriculture (e.g., Concentrated Animal Feeding Operations (CAFOs)), and extensive oil and gas (O&G) exploration in the Northern



Front Range contribute to the air pollution in the region. Recent studies have shown O&G emissions of non-methane hydrocarbons (NMHC) such as short-chain alkanes ($C_1$-$C_4$) and alkenes act as precursors to ozone (Pétron *et al.*, 2012; Edwards *et al.*, 2013; Gilman *et al.*, 2013; Karion *et al.*, 2013; Pétron *et al.*, 2014), but the potential for these emissions to contribute to primary and secondary OA in the region has not been investigated. Agricultural practices and power-plant operations in the greater Colorado region contribute to visibility impairment and ecosystem degradation, through formation of secondary nitrate and sulfate containing compounds (Williams and Tonnessen, 2000; Nanus *et al.*, 2003; Blett *et al.*, 2004; Burns, 2004; Boy *et al.*, 2008; Malm *et al.*, 2013; Mast and Ely, 2013; Thompson *et al.*, 2015b).

Since late 70's, several air quality studies in the Colorado Front Range have been carried out to identify emission sources and meteorological conditions affecting the air quality of the urban corridor. Meteorological influences were investigated in a field program sponsored by the U.S. EPA in 1973 known as the Denver Air Pollution Study (Russell, 1976), with a focus on hourly surface and elevated airflow measurements during air pollution episodes in Denver during the month of November. Results described occurrences of rapid dispersal of pollutants to the north-northeast of Denver due to strong winds and recurring reversal of winds, bringing aged pollutants back to the urban center. Additionally, the Denver Haze Study conducted in the winter of 1978-1979 by General Motors and the Motor Vehicle Manufactures Association (MVMA), and the 1987-88 Metro Denver Brown Cloud study provided objective apportionment to the observed brown cloud pollution over Denver. These studies evaluated the influence of the wintertime inversion layer and emission sources, including those of gas and coal burning from local power plants, on air quality and visibility degradation. Among the measured aerosol species, elemental carbon, ammonium sulfate, and ammonium nitrate contributed to the majority of optical extinction, decreasing visibility in the visible range by about 38%, 20%, and 17%, respectively (Countess *et al.*, 1980; Groblicki *et al.*, 1981; Wolff *et al.*, 1981; Watson *et al.*, 1988; Neff, 1989).

During 1996-1997, measurements of aerosol composition and inorganic aerosol precursors were carried out in winter and summer months at several urban and rural sites during the Northern Front Range Air Quality Study (NFRAQS). Summertime 24-h $PM_{2.5}$ mass concentrations at different sites ranged from 4-26 µg m$^{-3}$ while winter measurements indicated variable $PM_{2.5}$ mass in the range of 1-51 µg m$^{-3}$, depending on the sampling location and year (Watson *et al.*, 1998). During Summer 1996 and at an urban site northeast of downtown Denver, OA was the most dominant component of $PM_{2.5}$ mass, contributing to 46% of the mass with an average organic carbon mass of 4.2 µg m$^{-3}$ (Watson *et al.*, 1998). During this time, secondary inorganic aerosol contributed to 18% of $PM_{2.5}$ mass, about 50% lower than the wintertime observations, with average sulfate and nitrate concentrations of ~1.4-1.5 µg m$^{-3}$ and 0.9-1.2 µg m$^{-3}$, respectively (Watson *et al.*, 1998). On average, crustal components of $PM_{2.5}$ were low in concentration (less than 0.5 µg m$^{-3}$) during Summer 1997 (Watson *et al.*, 1998). Since the late 1990's, emissions in the Front Range have likely changed due to changes in the vehicular fleet, urbanization, and growth in O&G related activities. Despite these changes, recent comprehensive characterization of summertime air quality in the Colorado Front Range has been lacking.



The complex topography of the Colorado Front Range leads to terrain-induced flows and mesoscale circulations that have a significant impact on air quality. These include cycles of daytime thermally-driven upslope from the plains into the mountains and decoupled, downslope nighttime drainage and slope flows which can transport and pool particulates and precursors of secondary aerosols into the wider Platte River Valley between Denver and Greeley, Colorado. Thermally-driven upslope flows or cool moist northeasterly upslope flows can lead to secondary aerosol formation and poor visibility (Neff, 1997). Many of these upslope flows can be caused by low-pressure formation in southern Colorado, a lee trough or line of lower pressure along the foothills, and the Denver Cyclone. The Denver Cyclone (Wilczak and Glendening, 1988; Wilczak and Christian, 1990; Szoke, 1991; Szoke *et al.*, 2006) is a mesoscale cyclonic gyre which can form when there are southeasterly flows across the Palmer Divide (an east-to-west feature of higher terrain to the south of Denver) and a layer of high stability above the surface mixed layer and below 700 hPa (Szoke and Augustine, 1990; Reddy and Pfister, 2016). Reddy *et al.* (1995) have shown that the Denver Cyclone plays a key role in the degradation of visibility and exceedances of the state visibility standard of 76 Mm$^{-1}$ during the winter, but our study is the first to examine the impacts of the Denver Cyclone during an intensive air quality study with a detailed suite of aircraft and surface measurements.

In the summer of 2014, two major field campaigns, the Front Range Air Pollution and Photochemistry Éxperiment (FRAPPÉ) cosponsored by NSF/NCAR and the Colorado Department of Public Health and Environment (CDPHE), and the 4[th] deployment of the NASA DISCOVER-AQ, were carried out to study summertime atmospheric pollution in the Northern Colorado Front Range. In this manuscript, we focus our analysis on the data obtained during FRAPPÉ to assess the impact of the Denver Cyclone on the region's air quality (Flocke, 2015).

## 2 Measurements

### 2.1 Field Campaign

Airborne measurements were made during the Front Range Air Pollution and Photochemistry Éxperiment (FRAPPÉ) from 16 July-18 August 2014. Fifteen research flights were conducted over the Northern Colorado plains, foothills, and west of the Continental Divide to sample air masses under the influence of diverse sources and meteorological patterns that impact the overall air quality in the region. In this analysis, measurements made in the geographical area of the greater Denver Metropolitan area (latitudes of 39°27′00″- 40°15′36″ N and longitudes of 104°17′24″- 105°19′48″ W) and northern Colorado counties in the Northern Front Range (NFR) (latitudes of 40°15′58″– 41°00′00″N and longitudes of 104°45′00″- 105°19′48″W) during days when the Denver Cyclone was strongly developed (27-28 July) are contrasted to measurements made during days without the presence of a Denver Cyclone (26 July, 02-03 August). Airborne data presented in this analysis are limited to measurements in the boundary layer (i.e., altitudes below 2300 m east of the foothills as further discussed in Section 2.3) to capture air masses impacted by various local sources.



## 2.2 Instrumentation

*In-situ* size-resolved composition measurements of non-refractory submicron aerosols (NR-PM$_1$) (organic (OA), nitrate (NO$_3^-$), sulfate (SO$_4^{2-}$), ammonium (NH$_4^+$), and chloride (Cl$^-$)) were made with an aerosol mass spectrometer, equipped with a compact time-of flight detector (mAMS, Aerodyne Inc.). Principle details of the instrument are described in depth elsewhere (Jayne *et al.*, 2000; Drewnick *et al.*, 2005; Canagaratna *et al.*, 2007). In short, aerosols form a narrow particle beam by passing through an aerodynamic lens system (Liu *et al.*, 1995a, b). After travelling through the high-vacuum particle time-of-flight chamber and impacting on an inverted-cone tungsten vaporizer at approximately 600 ˚C, non-refractory components of aerosols are evaporated and ionized by electron impact ionization. The data are acquired in 15 s intervals in two distinct acquisition modes (Jimenez *et al.*, 2003). In the particle time-of-flight (PToF) mode, the particle beam is modulated by a multi-slit chopper system, allowing for particle sizing. In the mass spectrometry mode (MS), the chopper completely blocks or opens the particle beam, allowing the determination of the ensemble mass spectra of aerosol species.

Ambient aerosols were sampled through a secondary diffuser inside a forward facing NCAR High-performance Instrumented Airborne Platform for Environmental Research (HIAPER) modular inlet (HIMIL) (Rogers, 2011), mounted under the aircraft, with a total residence time of 0.5 s between the HIMIL inlet and the AMS. Assuming the sample flow reached the same temperature as the cabin air within this time, relative humidity of the sample flow was estimated to be less than 40% for the data presented here. For the ambient conditions in the boundary layer (i.e., 20 ˚C and 70 kPa), the secondary diffuser inlet was estimated to be a PM$_2$ inlet, i.e., with 50% transmission efficiency of 2 μm spherical particles (density of 1500 kg/m$^3$). A pressure controlled inlet (PCI) (Bahreini *et al.*, 2008) was used to maintain a constant pressure of 350 Torr in the AMS inlet to eliminate fluctuations in particle size and transmission efficiency with ambient pressure variations.

Measurements of gas-phase tracers used in this analysis include carbon monoxide (CO), measured by vacuum UV resonance fluorescence (Gerbig *et al.*, 1999; Holloway *et al.*, 2000; Takegawa *et al.*, 2001) on the C130 and by Differential Absorption Carbon Monoxide Measurement (DACOM) instrument with an in-situ diode laser spectrometer system (Choi *et al.*, 2008; Warner *et al.*, 2010) on the NASA DISCOVER-AQ P-3 aircraft. NO$_x$ (NO and NO$_2$), were measured by Chemiluminescence (Ridley *et al.*, 2004). Mixing ratios of NO$_y$ (total reactive oxidized nitrogen species) were estimated as the sum of NO$_x$, nitric acid (HNO$_3$) (Huey *et al.*, 1998; Huey, 2007), peroxyacetyl nitrate (PAN), and peroxypropionyl nitrate (PPN), all measured by chemical ionization mass spectrometry (CIMS) (Slusher *et al.*, 2004), and alkyl nitrates (ANs), measured using thermal dissociation-laser induced fluorescence (TD-LIF) (Thornton *et al.*, 2000; Day *et al.*, 2002). A compact Quantum Cascade Tunable Infrared Laser Differential Absorption Spectrometer (QC-TILDAS) was used for ammonia (NH$_3$) measurements (Ellis *et al.*, 2010), while C$_2$H$_6$ and CH$_2$O were measured by mid-infrared spectrometry using the Compact Atmospheric Multi-species Spectrometer (CAMS) (Weibring *et al.*, 2006; Weibring *et al.*, 2007; Richter



*et al.*, 2015). Volatile organic compounds (VOCs), including $C_6$-$C_9$ aromatics, were measured by online proton-transfer mass spectrometry (PTR-MS) (Lindinger *et al.*, 1998; de Gouw and Warneke, 2007).

### 2.3 Calibration and Data Processing

The mass response of the AMS was calibrated regularly by sampling size-selected, dry, monodisperse $NH_4NO_3$ particles with the procedure and calculations described in previous literature to determine ionization efficiency (IE) of nitrate and ammonium (Jimenez *et al.*, 2003; Zhang *et al.*, 2005). The average ratio of the nitrate ionization efficiency ratio to the air beam signal was $(2.57\pm0.26) \times10^{-13}$ from 5 calibrations performed during the study, indicating stability of the instrument throughout the project. Composition-dependent collection efficiency was applied to all the data in this study (Middlebrook *et al.*, 2012). AMS data analysis was carried out using the standard SQUIRREL analysis software (v1.56, (Sueper, 2015)) with Igor Pro 6.37 (WaveMetrics, Lake Oswego, OR).

Reported data are a subset of the FRAPPÉ 2014 data collected aboard the NSF/NCAR C-130 aircraft. All data presented here were limited to air masses sampled below ~2300 m ASL and values for aerosol concentrations are reported at STP (1013 hPa and 273 K, µg sm$^{-3}$). Additionally, data were chosen from days before (26 July), during (27-28 July), and after (02-03 August) the Denver Cyclone period, with the strongest features of the cyclone being observed on 27 July. To evaluate the impact of the Denver Cyclone in different regions of the Front Range, measurements were analyzed in three regions, labeled as In-Flow, Northern Front Range (NFR), and the Denver Metropolitan Area (DM), based on cluster analysis of wind patterns and aerosol and gas phase tracer concentrations observed on the day with the strongest Denver Cyclone, 27 July. Flight tracks and outlines of the latitude and longitudinal boxes for these regions are shown in Fig. 1.

To assess the extent of boundary layer mixing and dilution, potential temperature profiles measured by the Pennsylvania State University NATIVE integrated ozonesonde (Thompson *et al.*, 2015a), launched near Platteville (40°10'53" N, 104°43'36" W) during NASA DISCOVER-AQ, were examined. Except for 26 July when at 12:00 MST the boundary layer (BL) height was observed to be at 2200 m ASL, mid-day BL heights on other days were consistently at ~3400-3600 m ASL. Additionally, except for the high, constant-altitude legs, the sampling altitude on 26 July and the other flights were lower than ~2000 m ASL and ~2300 m ASL, respectively. Therefore, the data discussed here represent mainly those of the boundary layer air masses. Variability in the extent of boundary layer dilution due to differences in daytime flight hours (take-off times of 8:30- 14:00 MST) showed some effects on the observations; however, as further discussed in Section 3.3.1, dilution differences were not the main driving factor in the observed trends of absolute concentrations of gaseous and aerosol species.

### 2.4 ISORROPIA II Modeling

An aerosol thermodynamics model, ISORROPIA II (Nenes, 2013) was used to predict the phase and composition of the major inorganic aerosol components. Detailed equilibrium relations and thermodynamic parameters used in ISORROPIA II are outlined in Fountoukis and Nenes (2007). The model was initiated with the average measured values of temperature





(T), relative humidity (RH), and total concentrations of ammonium ($NH_{3\,(g)}$ + $NH_4^+$), sulfate ($SO_4^{2-}$), and nitrate ($HNO_{3\,(g)}$ + $NO_3^-$). Assuming chemical equilibrium and presence of metastable aerosols, the model predicted concentrations of sulfate, nitrate, and ammonium present in the gas and aerosol phase, allowing estimation of the aerosol nitrate fraction ($f_{NO3}$=$NO_3^-$ /($HNO_{3\,(g)}$+ $NO_3^-$)) at equilibrium.

## 3 Results and Discussion

### 3.1 Meteorology

Meteorological measurements presented in Table 1 show average ambient temperature (T), RH, and wind speed (WS) during selected flights for each of the three regions of interest on non-cyclone and cyclone days, respectively. During non-cyclone days, T, RH, and WS were similar in all regions with an average of 23 ± 1.6 ˚C, 35 ± 6.0%, and 3.4 ± 1.5 ms$^{-1}$, respectively. During the cyclonic episode, the average T across all three regions was 22 ± 1.6 ˚C and lower by 2-8% in the NFR and DM areas compared to the In-Flow region. Additionally, average RH was higher in NFR and DM (64-70%) compared to the In-Flow region (37%) during this mesoscale event. We further address the importance of the contrast in RH between the events for aerosol nitrate partitioning in Section 3.5. Average wind speed showed a 65% percent increase in the In-Flow region (6.3 ± 1.9 ms$^{-1}$) during the cyclone event, with a gradual decrease in the average wind speeds across the NFR and DM.

We used analysis runs of the National Centers for Environmental Prediction (NCEP) 13 km resolution Rapid Refresh (RAP) model for the periods of interest. These analysis runs reflect extensive assimilation of observational data. Plots were generated and analyzed with surface wind direction/speed and RH for days with and without the influence of the cyclone. Surface wind direction/speed for both case scenarios are shown in Fig. 2 and Fig. 3. As previously described by Toth and Johnson (1985), cyclic terrain-driven circulations in this region are common during the summer when synoptic-scale influences are weak. When synoptic scale flows are weak and the Denver Cyclone is not active, nighttime and early morning slope and drainage flows are formed as radiative cooling in the higher terrain to the north, west, and south of the DM causes denser, cooler air to flow downhill and with a general westerly component along the valleys over Denver (Fig. 2a and 2c). The surrounding terrain channels this drainage flow to the northeast through Denver. This flow can carry emissions away from the urban center. During the day, typical thermally-driven flows reverse these winds, and transport is generally towards the higher terrain. This daytime regime can also interact with synoptic scale winds leading to a hybrid pattern. Such a pattern is apparent for the daytime winds plotted in Fig. 2b and 2d, where thermally-driven upslope flow was more apparent over the higher terrain to the west, and synoptic-scale flows had a greater influence over the plains. Short-range return flows which can be formed by various mesoscale phenomena (Reddy et al., 1995), including the Denver Cyclone, can occur any time of the day and lead to a shift in direction of the winds with an easterly component. These can draw the Platte Valley air masses uphill and back over the greater Denver Metropolitan area, enhancing the mixing of older and new emissions (Neff, 1989).





Various tracers were considered in the Weather Research and Forecasting Model (WRF) to predict the distribution of emitted pollutants in the Front Range at a horizontal resolution of 3 km x 3 km. The model was initialized with the Global Forecast System (GFS) at 0.5°x 0.5° resolution and at 00:00 UTC (17:00 MST, on previous day) or 12 UTC (5:00 MST) to produce 48-h forecasts. Fig. S1 (a-f) of the supplementary material presents the distribution of the O&G tracer on 27 July.

These forecasting results represent the cyclone development on 27 July well, with the surface winds reflecting the counter-clockwise circulations (NE to SW) though the cyclone core was predicted to be further northeast of the Denver urban area. In this case, the model was able to predict the cyclone episode and transport of emission tracers 24 h in advance, driving the motivation for carrying out aircraft measurements during this event.

Pronounced and fully developed surface mesoscale circulations of the Denver Cyclone were observed on Sunday,
27 July 2014. Surface wind patterns and RH in Fig. 3 (a-d) display the development of the Denver Cyclone between 10:00 UTC and 18:00 UTC (3:00 MST and 11:00 MST, correspondingly) on 27 Jul. Fig. 3a depicts the early stages of the cyclone with converging flows and the beginnings of a counterclockwise circulation pattern centered to the northeast Denver. As seen in Fig. 3b, by 12:00 UTC (5:00 MST on 27 July), RH was beginning to peak on the western or return flow side of cyclone center which was still to the northeast of Denver. This northeasterly, northerly, northwesterly return flow on the
western side of the cyclone transported cool and moist air masses from the Platte Valley north of Denver towards the urban core as the cyclone matured. As seen in Fig. 3d, a well-organized, well-defined cyclone circulation continued with its center in the same location at 18:00 UTC (11:00 MST on 27 July) with a warm, dry inflow to the east of the center and convergence line and a cool, humid wrap-around flow on the west side of the Denver Cyclone. In the next sections, the impacts of this synoptic scale re-circulation flow on pollutant distribution in the region are discussed.

**3.2 Spatial distribution of trace gases and aerosols**

The meteorological conditions described above are critical when considering atmospheric aerosol formation, evolution, and spatial distribution. Fig. 4 (a-f) shows the spatial distribution of ammonia ($NH_3$), ethane ($C_2H_6$), and carbon monoxide (CO), i.e., tracers for agricultural and Concentrated Animal Feeding Operations (CAFOs), oil and gas exploration and production (O&G), and combustion and vehicular emissions, respectively, on non-cyclone and cyclone days.
Additionally, spatial representations of nitrogen oxides ($NO_x$), secondary gaseous pollutants ($O_3$ and PAN) and major aerosol components (OA, $NO_3^-$, and $SO_4^{2-}$) during non-cyclone and cyclone days are shown in Fig. 5 and Fig. 6.

Consistent with the meteorological conditions presented above, there is a stark contrast in the spatial distribution of pollutants during the non-cyclone and cyclone situations. Westward transport of emissions was seen on the non-cyclone (26 July, 02-03 August) days with the separation of pollutants in the northern and southern latitudes as depicted in Fig. 4 (a-b)
for $C_2H_6$ and $NH_3$. Ethane observations indicate that emissions from O&G, which are concentrated northeast of Denver, were mostly localized downwind and to the west of the sources during the non-cyclone periods. $NH_3$ point sources are predominantly concentrated in areas near Fort Collins and Greeley where a significantly large number of animal and livestock feeding operations reside. Nitrate production has both an urban and agricultural component due to oxidation of



$NO_x$ to $HNO_3$, subsequent reaction of $HNO_3$ with gas phase $NH_3$, and partitioning of $NH_3$ into the aerosol phase. These interactions will be explored further with ISORROPIA II model in Section 3.5. The cyclonic circulation on 27-28 July transported emissions from point sources in NFR down to DM (e.g., $C_2H_6$ and $NH_3$ in Fig. 4 d-e) and concentrated secondary photochemical products (e.g., $O_3$, PAN, OA, and $NO_3^-$ in Fig. 5e-f and Fig. 6d-e) in and around Denver/Boulder metropolitan compared to the northern counties (Fig. 4 d-e). Regional trends in trace gas and aerosol concentrations during cyclone and non-cyclone periods are discussed in Section 3.3.

### 3.3 Trends in trace gas and aerosol concentrations

Variations in spatial distribution of pollutants during the cyclone and non-cyclone events highlight the impacts of numerous sources and meteorology on air quality and aerosol formation within the Front Range. Here, we evaluate measurements of several auxiliary gases and aerosol chemical composition to gain insights on the influence of atmospheric dynamics on aerosol formation in the three regions of interest in the Front Range.

### 3.3.1 Gas-phase tracers

As discussed in Section 3.2., depending on the presence or absence of the cyclone, trace gases were transported and dispersed differently in the region. In Fig. 7 statistical distribution of several gas phase tracers, namely $NH_3$, $C_2H_6$, sum of $C_6$-$C_9$ aromatics, and CO measured in the In-Flow, NFR, and DM during the non-cyclone and cyclone periods are shown. Volatile organic compounds (VOCs) play important roles as atmospheric precursors to ground-level ozone and SOA (Turpin and Huntzicker, 1995; Song *et al.*, 2005; Volkamer *et al.*, 2006; Kroll and Seinfeld, 2008; Hallquist *et al.*, 2009; von Stackelberg *et al.*, 2013; Riva *et al.*, 2015) The aromatics highlighted in Fig. 7 represent a subset of the measured VOCs, typically found in O&G and vehicular emissions, that are known to form SOA (Ng *et al.*, 2007; Gentner *et al.*, 2012).

During the non-cyclone periods, the mean mixing ratio of $NH_3$ (Fig. 7a) in In-Flow and NFR areas was 13±11 ppbv while a significantly lower mean mixing ratio (3.8±3.6 ppbv) was observed in DM, owing to the high concentration of major ammonia point sources in the northeastern parts of the Front Range. Additionally, the mean mixing ratio of $C_2H_6$ (Fig. 7b) was higher by a factor of 2-2.6 in NFR (11.9±8.0 ppbv) compared to In-Flow (4.6 ±4.1 ppbv) and DM area (6.0±7.8 ppbv), due to substantial density of O&G exploration activities in NFR. For $\sum C_6$-$C_9$ aromatics (Fig. 7c), mixing ratios were higher over DM (~0.4-0.5 ppbv) compared to NFR (~0.15-0.3 ppbv) during both cyclone and non-cyclone events. This is in contrast to the pattern observed for $C_2H_6$, suggesting that the emission sources of $C_6$-$C_9$ aromatics are more concentrated in DM. Similar to $\sum C_6$-$C_9$ aromatics and consistent with combustion processes being the dominant source of aromatics and CO, mean mixing ratios of CO (Fig. 7d) were highest over DM during non-cyclone and cyclone periods.

Mean mixing ratios of CO over DM during the cyclone were 144±23 ppbv compared to 110±8.7 ppbv in In-Flow and 114±12 ppbv in NFR. Additionally, mean values of CO and $C_2H_6$ in DM increased during the cyclone events compared to non-cyclone days, by 13% and a factor of 2, respectively (Fig. 7b, 7d). Since vehicular sources of CO are concentrated in DM, the slight increase in CO over DM during the cyclone was likely due to changes in the background CO in the region





and a shallower morning boundary layer on 27-28 July. However, the increase in $C_2H_6$ could be due to release of emissions into a shallower morning boundary layer on cyclone days, the cyclonic mixing of air masses from northern latitudes with higher emissions of $C_2H_6$ from O&G operations, or a combination of these two phenomena. The observed increase in the mean $C_2H_6$ mixing ratio in DM during the cyclone compared to the non-cyclone days (10.2±6.2 ppbv vs. 6.0±7.8 ppbv,

respectively) was more significant than the observed increase for CO (i.e., 13%, 145±23.2 ppbv vs. 128±34.4 ppbv, respectively). These observations suggest that the significant increase in $C_2H_6$ mixing ratio observed over DM during the cyclone cannot be solely explained by BL height differences, but rather driven strongly by transport of $C_2H_6$-rich air masses from NFR into the DM. Similarly, cyclonic transport of $NH_3$ from the NFR to DM resulted in a 30% increase in average $NH_3$ mixing ratios over DM, from 3.8±2.8 to 8.8±3.9 ppbv while the mixing ratios in In-Flow and NFR did not change

significantly.

### 3.3.2 NR-Aerosol composition

Average boundary layer values of non-refractory submicron aerosol (NR-PM$_1$) composition in the Front Range on both non-cyclone and cyclone episodes are shown in Fig.8, with the exclusion of Cl$^-$ due to average mass loadings that were below its average detection limit of 0.19 μg sm$^{-3}$. Throughout the non-cyclone period, the average mass concentrations of

NR-PM$_1$ aerosols were consistently lower in all three regions, by a factor of ~2.5. Additionally, the NR aerosol was dominated by OA (75%, 3.25±1.45 μg sm$^{-3}$), followed by sulfate (13%, 0.58±0.27 μg sm$^{-3}$), ammonium (6%, 0.28±0.88 μg sm$^{-3}$), and nitrate (6%, 0.26±0.27 μg sm$^{-3}$) (Fig. 8a). During the Cyclone events, OA still dominated NR-PM$_1$ aerosol composition, but with a lower fraction (60%), while the contribution of nitrate, and correspondingly ammonium, increased to 16% and 11%, respectively. It is worth comparing the current measurements with those made during NFRAQS-Summer 96.

The overall composition of NR aerosols was similar in 1996, with OA as the dominant species present. However, assuming a conservative organic matter mass to organic carbon ratio of 1.7 (Turpin and Lim, 2001; Aiken *et al.*, 2008), OA mass of PM$_{2.5}$ during 1996 was estimated to be 7.14 μg m$^{-3}$, which is more than a factor of 2 higher than the average non-cyclone OA concentration during FRAPPÉ. Additionally, average concentrations of sulfate and nitrate during the NFRAQS-Summer 96 were factors of ~2-4 higher than those on the non-cyclone days of FRAPPÉ. Note that comparison of 1996 vs. 2014 data is

not exact due to higher (PM$_{2.5}$) size-cut of the 1996 measurements. During the winter Metro Denver Brown Cloud Air Pollution Study, aerosol composition was again dominated by OA (68%), followed by sulfate (14%), nitrate (10%), ammonium (8%), and chloride (<1%).

Shown in Fig. S2 are additional NR-PM$_1$ compositional pie charts for individual regions (In-Flow, NFR, DM) during the non-cyclone and cyclone periods of FRAPPÉ. As noted previously, OA was the single dominant species in all

three regions. Relative NR-PM$_1$ composition in In-Flow was most similar between the non-cyclone and cyclone periods whereas relative contribution of $NO_3^-$ increased during the cyclone period in NFR and DM at the expense of OA. Represented in Fig. 9 (a-c) are the observed trends in the NR-PM$_1$ aerosol concentrations (OA, $NO_3^-$, and $SO_4^{2-}$) measured in In-Flow, NFR, and DM during the non-cyclone and cyclone periods. Mass concentrations were consistently lower on non-



cyclone periods for all the measured aerosol species and within all three regions. On average, there was a 40% increase in average OA (Fig. 9a) on cyclone days across all three regions while the increase during the cyclone episode was up to ~80% for DM- an important consideration for air quality measures. During the non-cyclone days, average $NO_3^-$ was slightly higher in NFR ($0.43\pm0.39$ µg sm$^{-3}$) compared to DM ($0.20\pm0.20$ µg sm$^{-3}$), whereas during the cyclone episode, average $NO_3^-$ was a

factor of 3.3 higher in DM ($2.21\pm1.44$ µg sm$^{-3}$) compared to NFR ($0.67\pm0.54$ µg sm$^{-3}$). Overall, average $SO_4^{2-}$ (Fig. 9c) mass concentrations also displayed a 2-fold increase across all regions during the cyclone period. Consistent with the observations for $NH_3$ and $C_2H_6$, significantly larger increases in aerosol mass concentrations during the cyclone period were observed in DM compared to NFR, suggesting that mass concentrations during the cyclone were only slightly impacted by a shallower BL. Instead, transport of precursors and possibly aerosols from northern latitudes towards DM was the main driver for the

observed increased concentrations in DM. The fact that the highest aerosol concentrations during the cyclone period were observed in the greater DM underscores the importance of the impact of local meteorology on air quality in an area with a large population density.

**3.4 Photochemical Processing**

To assess the degree of atmospheric aging in air masses impacted by combustion, the relationship between primary

emitted $NO_x$ (sum of nitric oxide (NO) and nitrogen dioxide ($NO_2$)) and the resulting oxidized species $NO_y$ (sum of $NO_x+HNO_3+ANs+PAN+PPN$) was investigated. We utilized the ratio of $NO_x$ to $NO_y$, as a measure of photochemical processing of $NO_x$-containing air masses. As the ratio approaches one, the air masses are considered fresh while the value for the more aged air masses approaches zero (Kleinman *et al.*, 2007; DeCarlo *et al.*, 2008; Langridge *et al.*, 2012).

During the non-cyclone and cyclone periods, $NO_x/NO_y$ ratios were observed to be highest ($0.42\pm0.24$ and

$0.29\pm0.16$, respectively) over DM where freshly emitted plumes from vehicular traffic are dominant (Fig. 10). Further away from the urban center, $NO_x/NO_y$ values decreased with average values of $0.24\pm0.07$ in the In-Flow and NFR regions. Compared to the non-cyclone periods, during the cyclone events, $NO_x/NO_y$ values were similar in NFR while the average values decreased by 36% and 30% in In-Flow and DM regions, indicating further extent of photochemical processing of $NO_x$-containing air masses in these regions.

One caveat in this analysis may be the impact of lower $NO_x$ emissions during the weekends (26-27 Jul.), resulting in faster photochemistry and more secondary formation of $NO_y$ species and ozone. Several studies in high density population areas such as in Los Angeles have investigated the weekend effect on ambient ozone (Pollack *et al.*, 2012; Warneke *et al.*, 2013). These studies demonstrate that the higher ozone mixing ratios observed on weekends compared to weekdays is due to the significant weekend decrease in $NO_x$ emissions from diesel vehicles and a marginal, if any, decrease in the emissions of

non-methane hydrocarbons from gasoline vehicles, resulting in faster photochemistry, less ozone loss due to $NO_x$-titration, and more rapid ozone production (Pollack *et al.*, 2012; Warneke *et al.*, 2013). To examine changes in the weekend $NO_x$ emissions in the Front Range, we utilized the $NO_y$ and CO data measured in the boundary layer on-board the NASA P-3 aircraft during DISCOVER-AQ, which included data from a total of 8 weekday and 4 weekend flights from 17 July-10



August. During the weekends, $NO_x$ to CO enhancement ratio, determined by error-weighted (5% for $NO_x$ and 2% for CO) orthogonal-distance regression (ODR) fits, was lower by a factor of ~1.8 compared to weekdays (Fig. 11), which is in close agreement with observations made through aircraft measurements in the Los Angeles basin (Pollack *et al.*, 2012), indicating similar decrease in weekend diesel traffic in the Front Range as in Los Angeles.

In addition to the weekend change in photochemical processing of $NO_x$, the meteorological influence of a cyclone may also impact ozone, and possibly other secondary species, formation. Reddy and Pfister (2016) indicate that the Denver Cyclone is one of many "potential terrain-related mechanism for limiting area-wide dispersion of $O_3$ and its precursors". Trace gas spatial distribution maps, provided in Fig. 4-5, indeed indicated strong accumulation of secondary pollutants during the cyclonic event. Further analysis to investigate the impact of the cyclone on ozone formation in the Front Range

requires chemical box or regional modeling and is beyond the scope of this manuscript.

     Evolution of OA through photochemical aging during the cyclone and non-cyclone periods was studied in air masses with $NO_x/NO_y <0.5$, which represent intermediate to strongly processed $NO_x$-containing plumes. As the plumes age, an increase in the observed $\Delta OA/\Delta CO$ ratio suggests SOA production. In this analysis, we evaluated air masses sampled over DM to determine the extent of photochemical aging effects on Denver's local air quality. The error-weighted (30%

uncertainty in OA, 3% uncertainty in CO) linear ODR fits to the scatter plots of measured OA against background subtracted CO were obtained, with the slopes representing the ratios of $\Delta OA/\Delta CO$ (Fig. 12). Background CO values (90 ppbv and 110 ppbv during the non-cyclone and cyclone days, respectively) were based on the modes of the Gaussian curves fitted to the frequency distribution plots of CO. Uncertainties in the slopes represent the propagated uncertainties, i.e., the square-root of the quadric sum of the relative uncertainties in the ODR fit, OA concentration, and CO mixing ratio. The average cyclone

$\Delta OA/\Delta CO$ values were higher ($0.076\pm0.016$ µg sm$^{-3}$ ppbv$^{-1}$, r=0.59) compared to the non-cyclone periods ($0.046\pm0.017$ µg sm$^{-3}$ ppbv$^{-1}$, r=0.45), although not significantly considering the uncertainties associated with the fits. However, a significantly higher intercept of the fit was obtained on the cyclone days ($3.70\pm0.28$ µg sm$^{-3}$) compared to the non-cyclone days ($0.93\pm0.33$ µg sm$^{-3}$), indicating transport of additional OA relative to CO from the northern latitudes towards DM during the cyclone events. From an air quality standpoint, such enhancement in total OA concentration is significant since it is

comparable in magnitude to the average OA over DM during the typical non-cyclone summer days.

### 3.5 Aerosol nitrate production

     We assess the regional formation of aerosol nitrate through comparisons of aerosol nitrate fraction ($f_{NO3}=NO_3^-/(NO_3^- + HNO_3)$) in the In-Flow, NFR, and DM regions with and without the cyclone influence (Fig. 13a). Low $f_{NO3}$ values observed in the NFR and DM regions during the non-cyclone days indicate that nitric acid was predominantly present in the

gas phase. In contrast, higher $f_{NO3}$ values observed during the cyclone suggest increased partitioning of nitric acid to the condensed phase. As noted earlier, environmental factors including relative humidity, temperature, and atmospheric dynamics play important roles in the formation of aerosol nitrate (Stelson *et al.*, 1979; Stelson and Seinfeld, 1982; Watson, 2002). Slightly lower temperature and increased RH were observed in the NFR and DM during the cyclone period (Table 1).



Higher RH may enhance formation of nitrate aerosols by promoting aqueous and heterogeneous phase reactions and increasing the equilibrium partitioning of gas phase $NH_3$ and $HNO_3$ to the condensed particle phase (Stelson *et al.*, 1979; Stelson and Seinfeld, 1982; Volkamer *et al.*, 2006; Na *et al.*, 2007; Hessberg *et al.*, 2009). Moreover, local meteorology during the cyclone period, facilitating transport of $NH_3$ from the nearby feedlots in NFR to DM (Section 3.3.1, Fig. 4b,e),

could have favored equilibrium partitioning of nitric acid to the aerosol phase due to abundance of gas phase $NH_3$.

To further investigate the role of atmospheric conditions and mixing patterns in aerosol nitrate formation during the cyclone days, nitrate partitioning was evaluated by ISORROPIA II (Fountoukis and Nenes, 2007) model calculations, described in Section 2.4. The predicted partitioning results, summarized in Table S1 and Fig. 13a are in reasonable agreement with the observed $f_{NO3}$ values on non-cyclone and cyclone days. Over DM, the model predicted 24% more nitrate

existing in the aerosol phase compared to mean value based on the measurements; however, the predicted $f_{NO3}$ is still within the limits of variability of the observed $f_{NO3}$. To evaluate the influence of RH and T on aerosol nitrate formation, we considered model input variables based on the non-cyclone concentrations while prescribing the higher RH and lower T values representing conditions of the cyclone period (Table S1). In this case, the model predicted similar $f_{NO3}$ values in NFR and significantly lower $f_{NO3}$ over DM compared to the measurements, indicating that the observed higher partitioning of

nitrate to the aerosol phase during the cyclone events was not mainly driven by changes in ambient T and RH, but it was rather due to increased availability of $NH_3$ over DM with the cyclonic transport from NFR.

We further evaluated the influence of sulfate concentrations and ambient RH to understand how chemical composition and environmental changes in DM could impact nitrate partitioning between gas and aerosol phase (Table S2). While keeping T, RH, gas phase ammonia and ammonium associated with nitrate at the same level as in the baseline (i.e.,

observations on cyclone days over DM), the absence of aerosol sulfate results in a drastic increase in $f_{NO3}$, with almost all of the nitric acid partitioning to the aerosol phase. This result indicates that background aerosol sulfate concentrations have a strong effect on equilibrium partitioning of nitric acid. Next, we evaluated influence of RH, keeping all other variables the same as in the baseline. Increasing RH from 64% to 85% resulted in an increase in $f_{NO3}$ from 0.36 to 0.74 while decreasing RH to 35% decreased $f_{NO3}$ by a factor of 3.6. Taken together, these case scenarios suggest that meteorological transport

patterns, background sulfate concentrations, and RH all have significant influences on the phase equilibrium of nitric acid and aerosol nitrate formation. Although Denver metropolitan is not typically in violation of the $PM_{2.5}$ standard during summer months, higher aerosol nitrate concentrations may be observed in the presence of a cyclone and with RH values higher than what was observed during this study.

### 3.6 Impacts on optical extinction

Several studies have discussed the importance of nitrate containing aerosols on optical extinction ($\beta_{ext}$) coefficients, i.e., scattering and absorption of light, that impede visibility in affected regions (Tang, 1996; Watson, 2002; Li *et al.*, 2009; Langridge *et al.*, 2012; Zhang *et al.*, 2012; Lei and Wuebbles, 2013). As seen in Fig. 13b, average $\beta_{ext}$ values measured during FRAPPÉ ($\lambda=632$ nm) were similar in In-Flow, NFR, and DM region during non-cyclone days with an average of



10.6±3.5 Mm$^{-1}$, whereas factors of 1.5-3 increase in the average $\beta_{ext}$ were observed during the cyclone periods, with the most significant impact observed over DM.

Mass extinction efficiency values (MEE), defined as the slopes of the error-weighted (10% for $\beta_{ext}$, 30% for NR-PM$_1$ mass) linear ODR fits of $\beta_{ext}$ against total NR-PM$_1$ mass, were compared in Fig. 14. MEE values under the non-cyclone events in NFR and DM were 1.92±0.62 m$^2$ g$^{-1}$ (r=0.71) and 2.72 ± 0.61 m$^2$ g$^{-1}$ (r=0.62), respectively, higher by 42% in the urban center. During the cyclone events, MEE values were 43% higher over DM compared to NFR (2.85±0.63 m$^2$ g$^{-1}$ (r=0.84) and 1.99±0.64 m$^2$ g$^{-1}$ (r=0.88), respectively), but similar to the percentage increase observed during the non-cyclone days. On cyclone days a significant increase in the average mass concentrations of the aerosol species was noted (Fig. 8). However, similarity of MEE percentage increase in DM during the cyclone and non-cyclone days suggests that the increase in NR-PM$_1$ mass during the cyclone accompanied a similar increase in $\beta_{ext}$ and that MEE alone cannot provide detailed insights on the impact of the cyclone on $\beta_{ext}$ in DM.

As mentioned previously, the State of Colorado visibility standard has set a threshold of 76 Mm$^{-1}$ averaged over a 4 h period when RH<70%. To more directly investigate how the Denver Cyclone impacted visibility in DM, we refer to the CDPHE LPV-2 long-path transmissometer measurements of ambient $\beta_{ext}$ at 550 nm in downtown Denver during 11:00-15:00 MST (Table 2). During the non-cyclone days (26 July, 02-03 August), the 4 h average values $\beta_{ext}$ (550 nm) were 33-62 Mm$^{-1}$, well below the visibility standard. However, during the cyclone days (27-28 July), 4 h average $\beta_{ext}$ (550 nm) values were 90-139 Mm$^{-1}$, up to a factor of ~2 higher than the standard, resulting in *poor* ratings with respect to the visibility standard index (VSI).

To further understand the role of different aerosol components in driving the observed increase in airborne measurements of $\beta_{ext}$ (632 nm), correlations between $\beta_{ext}$ (632 nm) and NO$_3^-$, OA, and SO$_4^{2-}$ mass under the influence of non-cyclone and cyclone air masses were examined (Fig. 15). During the non-cyclone events, $\beta_{ext}$ displayed strong correlations (r=0.71) with OA and NO$_3^-$ in NFR and only with OA (r=0.70) in DM. $\beta_{ext}$ was poorly correlated with sulfate aerosols in the region during the non-cyclone events (r = -0.18, 0.11, for NFR and DM, respectively). During the cyclone events, all aerosol components equally influenced $\beta_{ext}$ in the NFR (r=0.88, 0.84, 0.87), while only strong correlations with NO$_3^-$ (r=0.86) were observed in DM. These results indicate that the Denver cyclone directly influenced visibility in the DM by facilitating transport of an additional aerosol precursor (i.e., NH$_3$) to the region compared to the non-cyclone events (detailed analysis in Section 3.5).

# 4 Conclusions

Data from FRAPPÉ-2014 project in the Colorado Front Range were presented to understand the influence of the Denver Cyclone on source distribution and processes that impact regional air quality and visibility in the summer. The analysis demonstrated that mesoscale re-circulation patterns changed the spatial distribution of pollutants emitted in the northern latitudes of the study area, transporting pollutants over the Denver Metropolitan (DM) area, leading to enhanced



concentration of secondary aerosol species. Overall, particle formation and growth during the non-cyclonic episodes occurred downwind of the major point/area sources and in isolation. Cyclonic transport from the In-Flow to the NFR forced air masses with a higher concentration of trace gases towards Denver, explaining the increased mixing ratios observed in DM. Average DM concentrations of OA and nitrate increased by ~79% and a factor of 10, respectively, during the cyclone

episodes.

The cyclonic flow facilitated transport of additional OA relative to CO from the northern latitudes towards DM, as seen by the increase in OA background compared to the non-cyclone days. Observations showed that the MEE values in the DM were similar under cyclone and non-cyclone days despite having different species (OA during non-cyclone and $NO_3^-$ during cyclone periods) driving $\beta_{ext}$ (632 nm). During the cyclone events, as confirmed by ISORROPIA II modeling and

ground-based measurements of optical extinction, summertime visibility in the Front Range was significantly impacted by the increase in aerosol nitrate formation due to abundance of $NH_3$ transported from the NFR region.

Overall, results from this study improve our understanding of sources and atmospheric processes responsible for summertime formation of aerosols in the greater Front Range and the impacts on air quality and regional haze during cyclone and non-cyclone events. Based on these results, reduction in source strengths of aerosol precursors in NFR leading

to OA and ammonium nitrate formation, including mitigation of $NH_3$ emissions from dairy and livestock farming, could dramatically reduce the impact of cyclone events on Denver's air quality and visibility.

**Acknowledgements**

K.T.V was partially supported by the NIEHS, T32 Research Training in Environmental Toxicology Grant. The authors would like to recognize Daniel Adams and Michael Fournier (University of California, Riverside CNAS machine shop) for

their craftsmanship, NCAR Research Aviation Facility (RAF) technicians for their support during integration and throughout the field campaign, Joshua Schwarz (NOAA-ESRL) for providing the secondary diffuser inlet system, Charles A. Brock (NOAA-ESRL) for providing the condensation particle counter for field calibrations, Ron Cohen and Carly Ebben (University of California, Berkeley) for TD-LIF data, and A.M. Thompson and W. Brune (NASA/Goddard and Penn State) for sonde data from the Penn State NATIVE sampling trailer. Project support and funding was from the Colorado

Department of Public Health and Environmental (CDPHE) as well as the USDA National Institute of Food and Agriculture, Hatch project Accession No. 233133. Data used in this analysis may be obtained at: http://www-air.larc.nasa.gov/cgi-bin/ArcView/discover-aq.co-2014?C130=1.

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



**Figures**

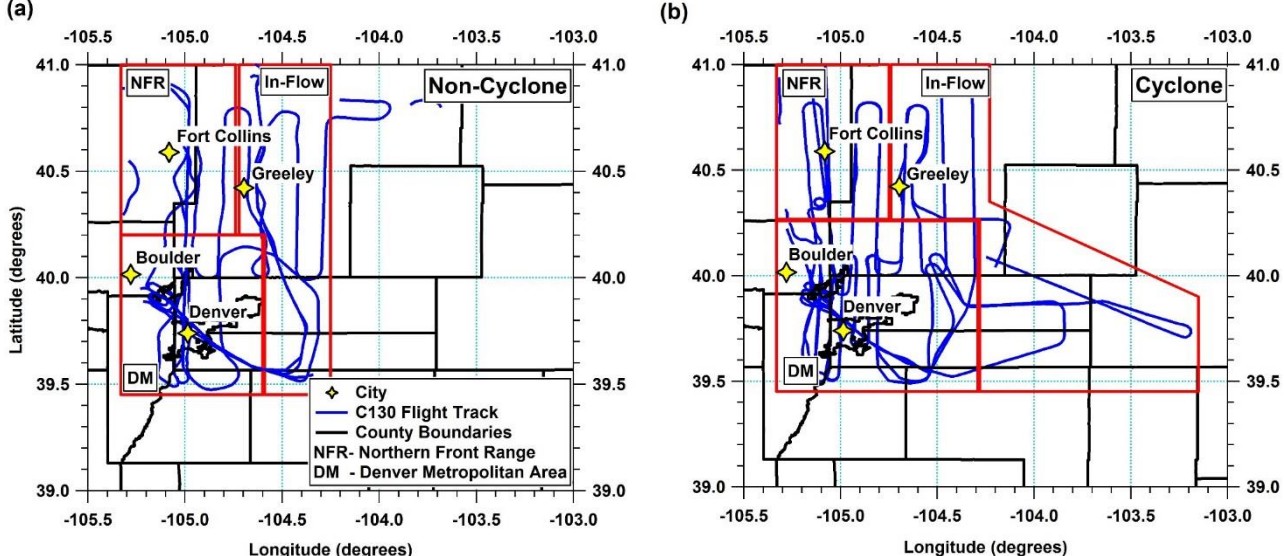

**Figure 1.** C-130 flight tracks in the Colorado Front Range for **(a)** non-cyclone days: Jul. 26, Aug. 02-03, 2014 and **(b)**
cyclone days: Jul. 27-28, 2014; red marked boundaries represent three different study regions: In-Flow, Northern Front
Range (NFR), and Denver Metropolitan Area (DM).





**Figure 2.** RAP model analysis runs at 13 km resolution for **(a-b)** Jul. 26, 2014 (12:00 UTC (5:00 MST), 21:00 UTC (14:00 MST), respectively) and **(c-d)** Aug. 02, 2014 (12:00, 21:00 UTC, respectively). Arrows show surface wind vectors while the color scale represents surface RH.



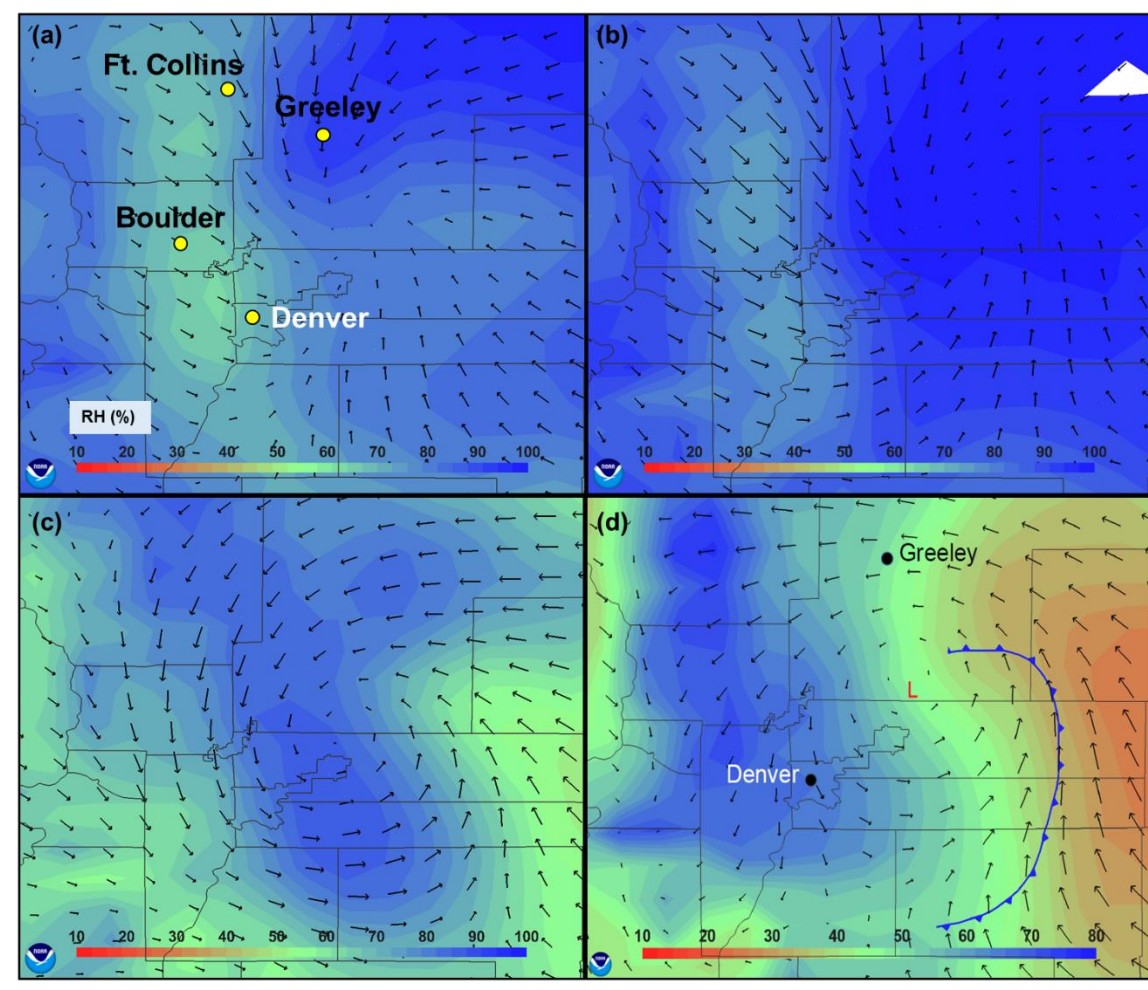

**Figure 3.** RAP model analysis runs at 13 km resolution for the Denver Cyclone on Sunday, Jul. 27, 2014 at **(a)** 10:00 UTC (3:00 MST), **(b)** 12:00 UTC (5:00 MST), **(c)** 15:00 UTC (8:00 MST), and **(d)** 18:00 UTC (11:00 MST). The *blue* line represents a convergence zone or front associated with the cyclone. Arrows show surface wind vectors while the color scale represents surface RH.





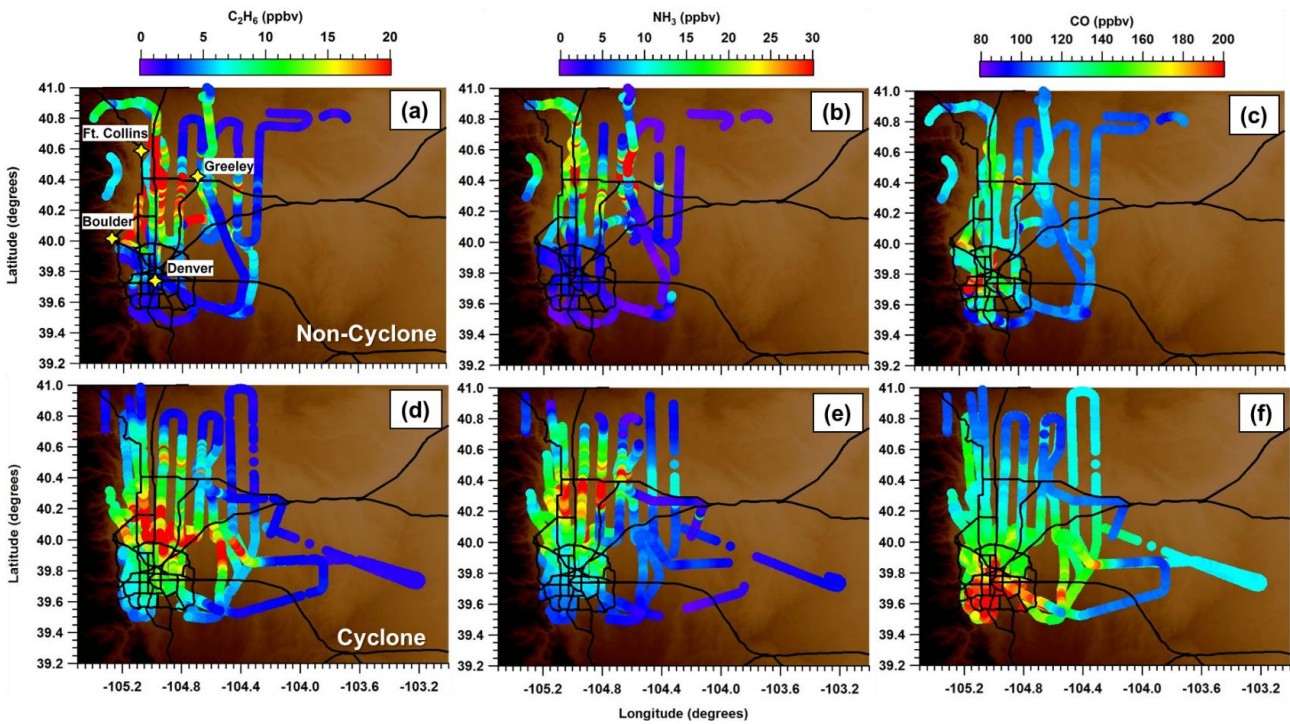

**Figure 4.** Spatial distribution maps of ethane ($C_2H_6$), ammonia ($NH_3$), and carbon monoxide (CO) in the Colorado Front Range during non-cyclone (a-c) and cyclone episodes (d-f). Major highways are shown with black lines.



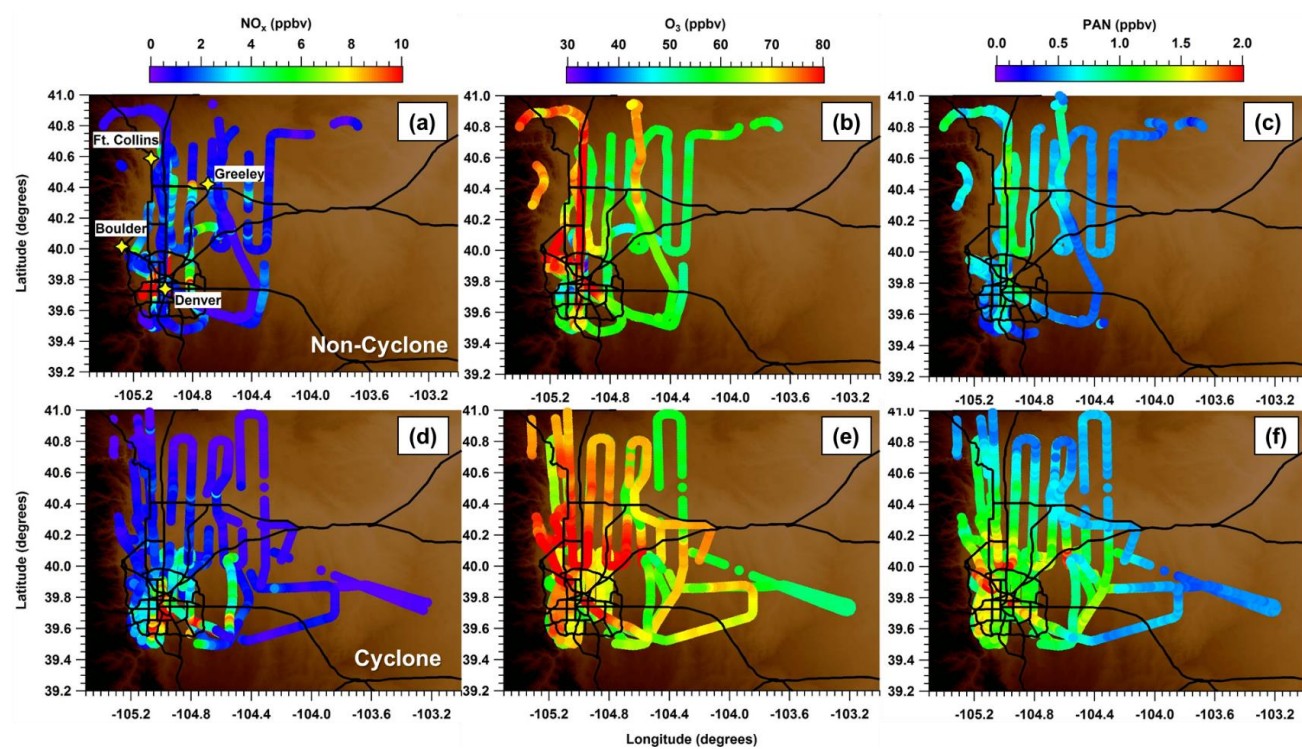

**Figure 5.** Spatial distribution maps of NO$_x$ [NO+NO$_2$], ozone (O$_3$), and peroxyacetylnitrate (PAN) in the Colorado Front Range during the non-cyclone (a-c) and cyclone episodes (d-f). Major highways are shown with black lines.





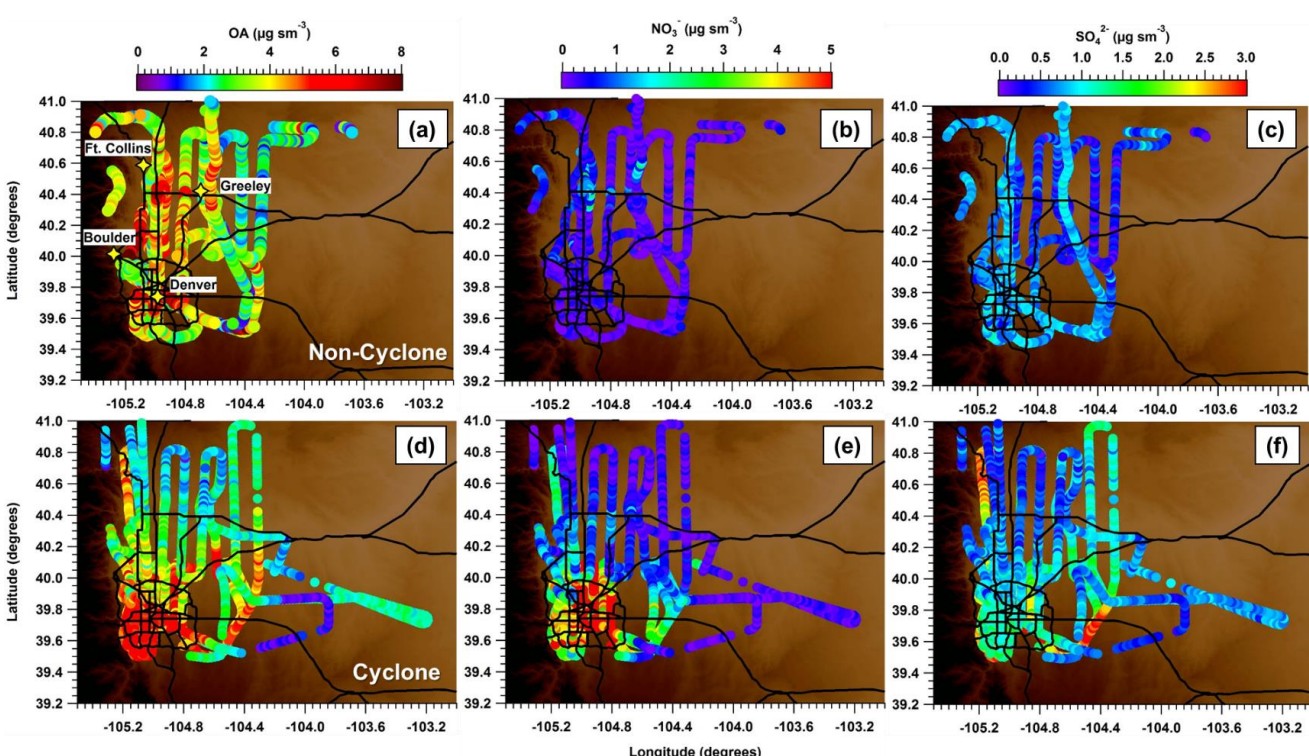

**Figure 6.** Spatial distribution maps of aerosol species (OA, $NO_3^-$, and $SO_4^{2-}$) in the Colorado Front Range during the non-cyclone (a-c) and cyclone episodes (d-f). Major highways are shown with black lines.





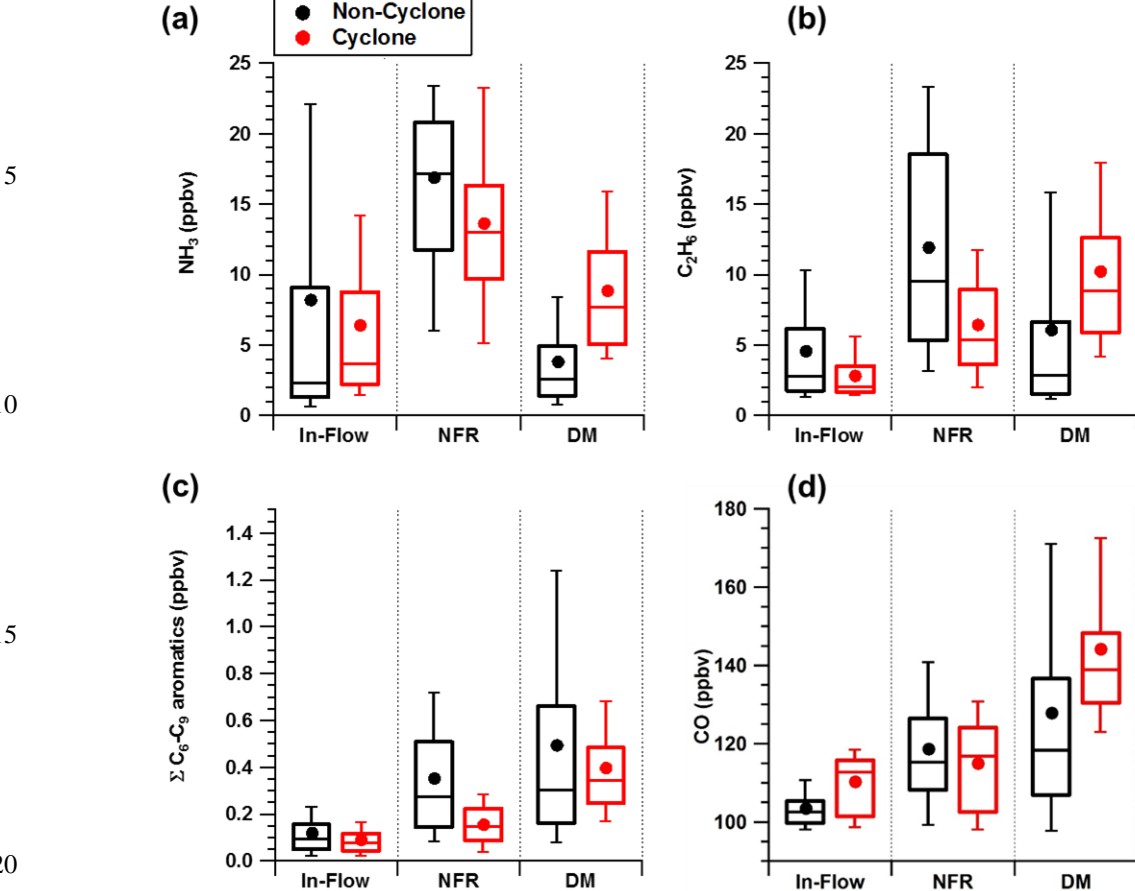

**Figure 7.** Statistical representation of the distribution of gas tracers ($NH_3$, $C_2H_6$, $\sum C_6$-$C_9$ aromatics, and CO) within the three study regions. The box and whiskers indicate $10^{th}$, $25^{th}$, $75^{th}$, and $90^{th}$ percentiles while the solid lines and circles mark the median and mean values, respectively.





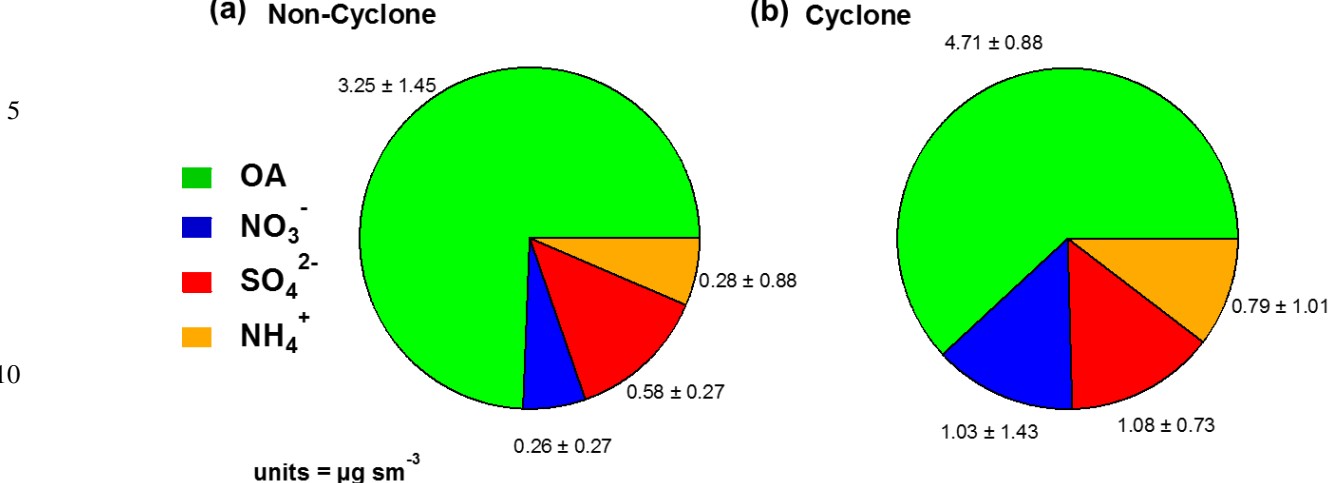

**Figure 8.** Average chemical composition of AMS species in all regions during **(a)** non-cyclone and **(b)** cyclone events. Chloride (Cl⁻), not shown, was below the instrument detection limit.





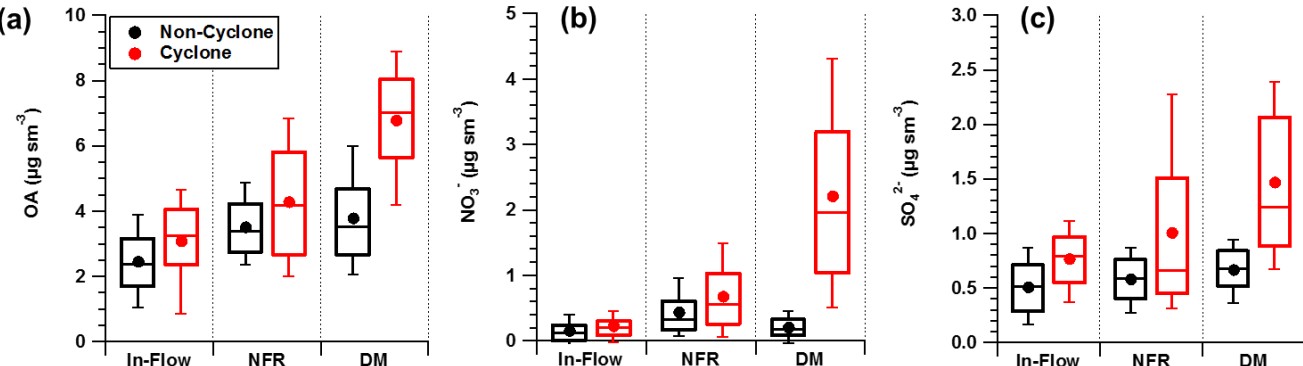

**Figure 9.** Statistical representation of the distribution of the mass concentrations of aerosol species (OA, $NO_3^-$, $SO_4^{2-}$) within the three study regions. The box and whiskers indicate 10th, 25th, 75th, and 90th percentiles while the solid lines and circles mark the median and mean values, respectively.





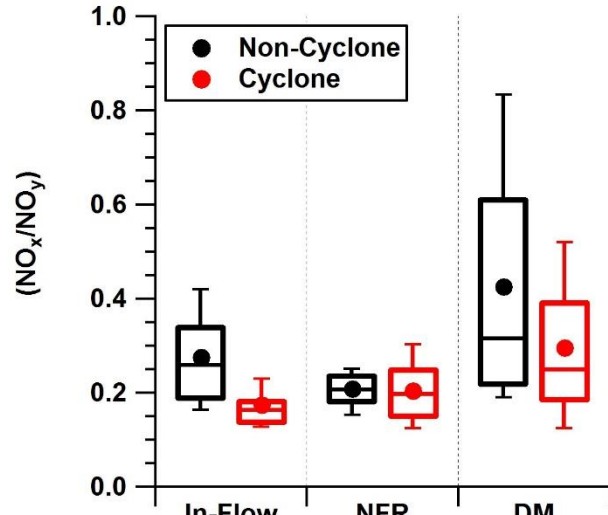

**Figure 10.** Statistical representation of the distribution of the mixing ratios of $NO_x/NO_y$ within the three study regions. The box and whiskers indicate $10^{th}$, $25^{th}$, $75^{th}$, and $90^{th}$ percentiles while the solid lines and circles mark the median and mean values, respectively.





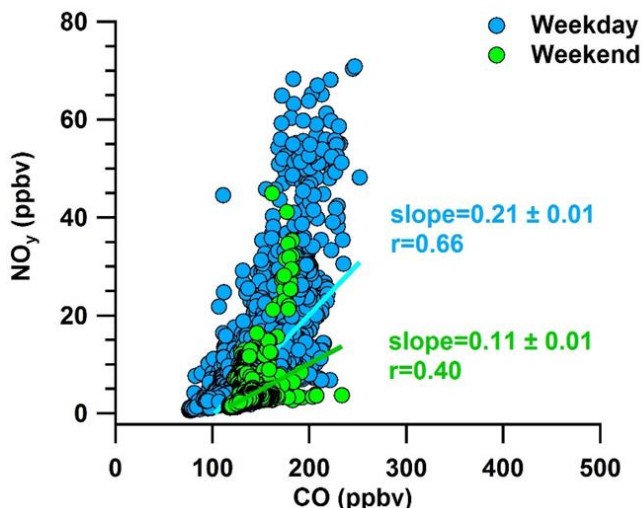

**Figure 11.** Scatter plot of measured $NO_y$ versus CO using aircraft data from the DISCOVER-AQ P-3 flights. Weekday (*blue* dots, 8 combined days) and weekend (*green* dots, 4 combined days). Inferred slopes are derived from ODR error weighted (5% $NO_y$, 2% CO) fits.





| DM, $NO_x/NO_y$ <0.5 | | | |
|---|---|---|---|
| **Scenario** | **Slope** | **y-intercept** | **r** |
| **Non-Cyclone** | $0.046 \pm 0.017$ | $0.93 \pm 0.33$ | 0.45 |
| **Cyclone** | $0.076 \pm 0.016$ | $3.70 \pm 0.28$ | 0.59 |

**Figure 12**. Scatter plot of OA ($\mu g\ sm^{-3}$) vs. $\Delta CO$ (ppbv) under the most aged air masses ($NO_x/NO_y$<0.5) in the DM for non-cyclone (*black*) and cyclone (*red*) days. Slope and intercept values are based on the ODR error weighted (30% OA, 3% CO) fits while the correlation coefficients are based on the linear least-squared regression fits.





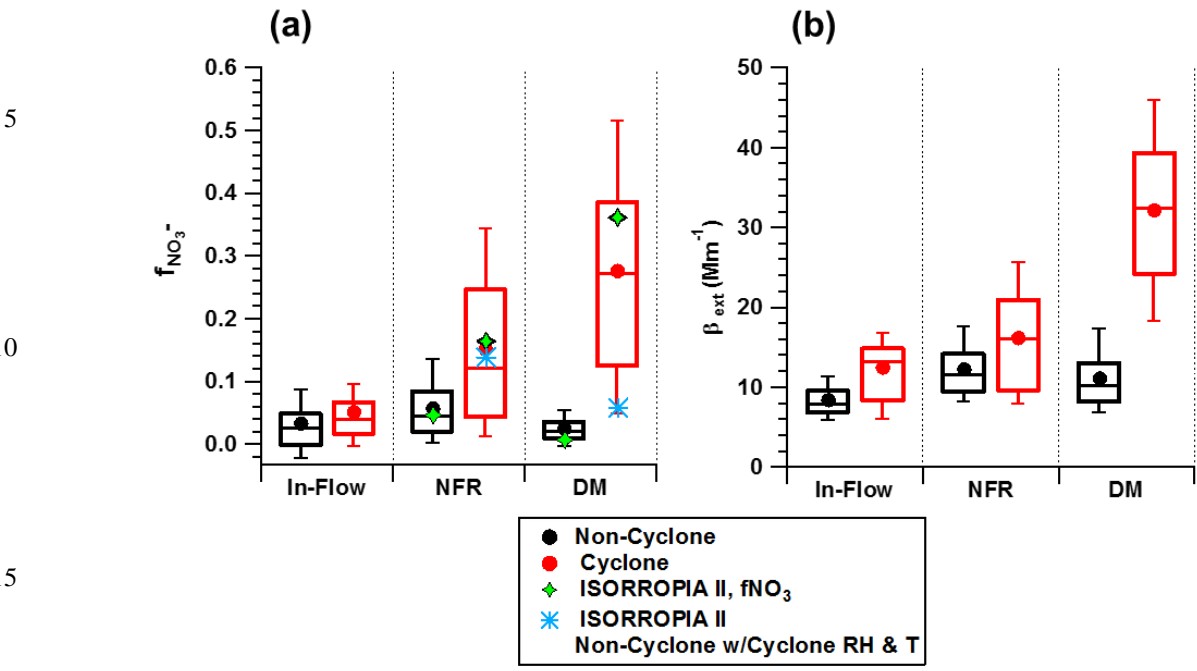

**Figure 13.** Statistical representation of the distribution of **(a)** aerosol nitrate fraction ($f$NO$_3$ = NO$_3^-$/ [NO$_3^-$+HNO$_3$]) and **(b)** aerosol optical extinction within the three studied regions during non-cyclone and cyclone periods. The box and whiskers indicate $10^{th}$, $25^{th}$, $75^{th}$, and $90^{th}$ percentiles while the solid lines and circles mark the median and mean values, respectively. Modeled $f$NO$_3$ values with actual inputs of chemical composition and T and RH are shown with *green* diamonds while the predicted values with the non-cyclone composition and cyclone T and RH are shown with *blue* stars.





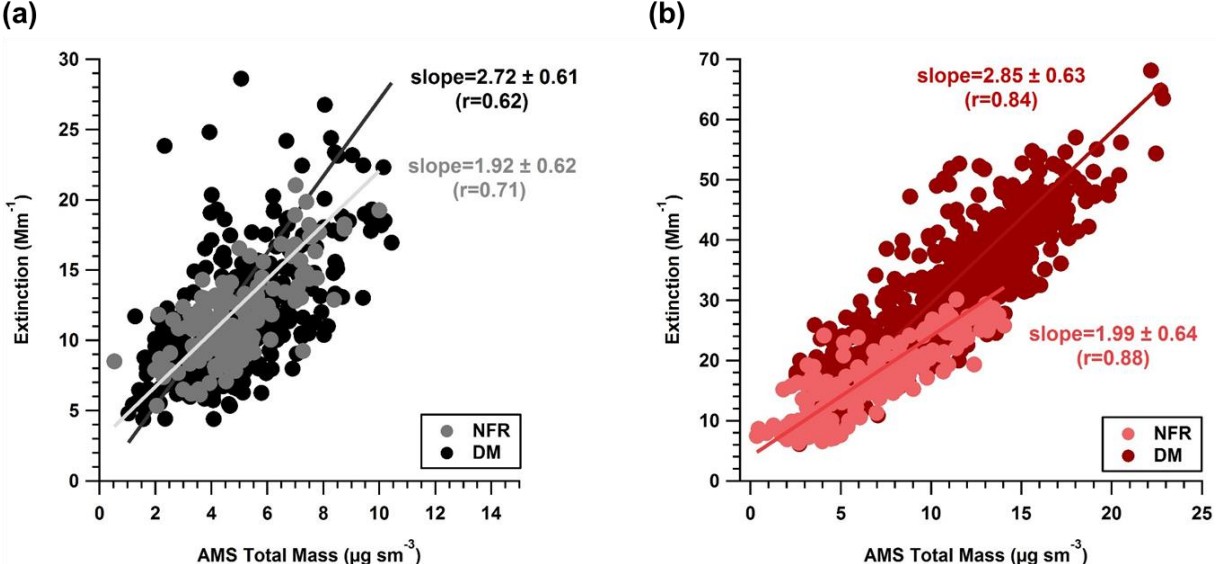

**Figure 14.** Mass extinction efficiency plots of $\beta_{ext}$ against total NR-PM$_1$ mass for NFR and DM during **(a)** non-cyclone and **(b)** cyclone episodes. Inferred slopes are derived from ODR error weighted (10% $\beta_{ext}$, 30% AMS Total Mass) fits.





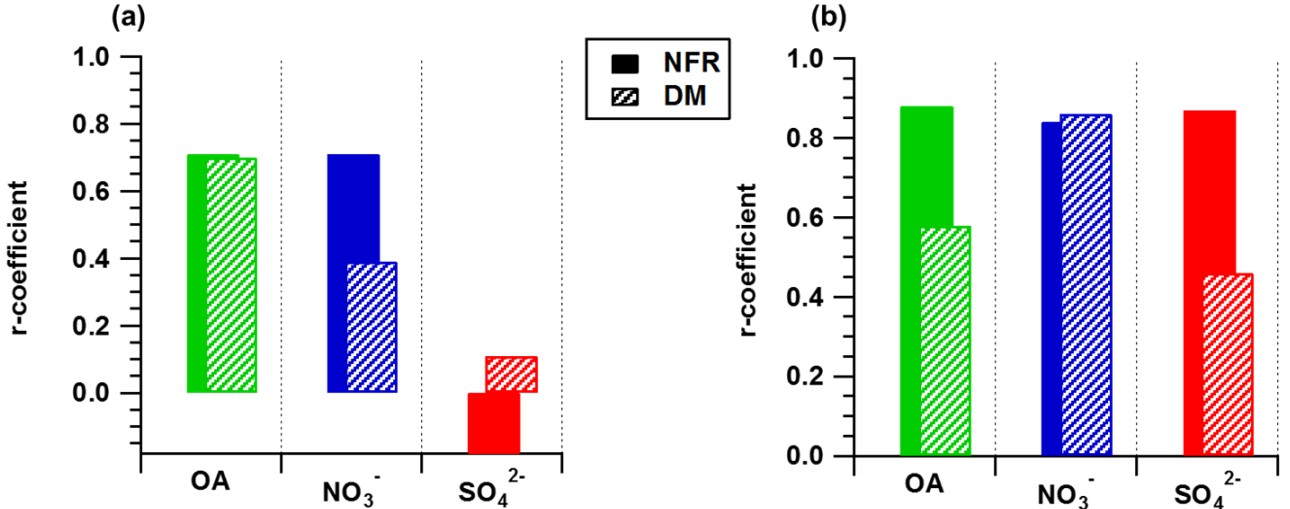

**Figure 15.** Correlation coefficients of scatter plots of $\beta_{ext}$ against individual aerosol species for NFR and DM during **(a)** non-cyclone and **(b)** cyclone episodes.





| Non-Cyclone (Jul. 26, Aug. 02-03) | | | |
|---|---|---|---|
| Region | T (˚C) | RH (%) | WS (m/s) |
| In-Flow | 22.8 ± 1.7 | 33.3 ± 4.6 | 3.8 ± 1.4 |
| NFR | 22.5 ± 1.7 | 38.4 ± 6.9 | 3.5 ± 1.6 |
| DM | 23.7 ± 1.4 | 34.0 ± 6.6 | 3.0 ± 1.3 |
| Cyclone (Jul. 27-28) | | | |
| In-Flow | 22.4 ± 1.4 | 37.0 ± 5.5 | 6.3 ± 1.9 |
| NFR | 21.8 ± 1.3 | 70.4 ± 7.2 | 4.1 ± 1.4 |
| DM | 20.6 ± 2.0 | 64.5 ± 7.7 | 3.2 ± 1.4 |

**Table 1.** Average temperature (T, ˚C), relative humidity (RH, %), and wind speed (WS, ms[-1]) for measurements separated into "In-Flow", "NFR", and "DM" regions during the non-cyclone and cyclone episodes.



| Date | Hour (MST) | $\beta_{ext}$ (Mm$^{-1}$) | RH (%) | 4 h Avg. (Mm$^{-1}$) | VSI | Descriptor |
|---|---|---|---|---|---|---|
| 26 Jul. 2014 | 11:00 AM | 70 | 33 | 59 | 66 | Moderate |
| | 12:00 PM | 50 | 29 | 59 | 66 | Moderate |
| | 1:00 PM | 50 | 30 | 57 | 64 | Moderate |
| | 2:00 PM | 51 | 32 | 55 | 60 | Moderate |
| | 3:00 PM | 45 | 33 | 49 | 49 | Good |
| 27 Jul. 2014 | 11:00 AM | 124 | 66 | 139 | N/A | N/A |
| | 12:00 PM | 108 | 61 | 125 | N/A | N/A |
| | 1:00 PM | 95 | 53 | 113 | N/A | N/A |
| | 2:00 PM | 110 | 49 | 109 | 145 | Poor |
| | 3:00 PM | 112 | 47 | 106 | 141 | Poor |
| 28 Jul. 2014 | 11:00 AM | 118 | 56 | 118 | N/A | N/A |
| | 12:00 PM | 108 | 51 | 112 | 149 | Poor |
| | 1:00 PM | 84 | 46 | 104 | 138 | Poor |
| | 2:00 PM | 78 | 43 | 97 | 129 | Poor |
| | 3:00 PM | 88 | 43 | 90 | 119 | Poor |
| 02 Aug. 2014 | 11:00 AM | 38 | 33 | 44 | 43 | Good |
| | 12:00 PM | 37 | 31 | 42 | 40 | Good |
| | 1:00 PM | 33 | 23 | 38 | 35 | Good |
| | 2:00 PM | 29 | 21 | 34 | 30 | Good |
| | 3:00 PM | 32 | 21 | 33 | 28 | Good |
| 03 Aug. 2014 | 11:00 AM | 53 | 40 | 62 | 72 | Moderate |
| | 12:00 PM | 44 | 34 | 57 | 63 | Moderate |
| | 1:00 PM | 41 | 29 | 50 | 50 | Good |
| | 2:00 PM | 37 | 25 | 44 | 42 | Good |
| | 3:00 PM | 37 | 24 | 40 | 37 | Good |

**Table 2.** Data summary table of $\beta_{ext}$ measurements from the CDPHE long-path transmissometer in downtown Denver for each of the 5 days of interest. On the Visibility Standard Index Scale, a value of 101 equates to 76 Mm$^{-1}$ standard. Values between 0-50 are described as good, 51-100 moderate, 101-200 poor and 201-plus extremely poor visibility.