# Peer review of "Impacts of the Denver Cyclone on Regional Air Quality and Aerosol Formation in the Colorado Front Range during FRAPPÉ 2014"

_Atmospheric Chemistry and Physics, 2016_

## Referee Comment (RC1) · Anonymous Referee #1 · 14 Jul 2016

The discussion paper describes measurements of gas phase and aerosol species over Colorado, and relates them to the Denver Cyclone. In general the measurements and analysis are good and well-described. The clarity of the presentation could be improved in several ways before final publication.

General comments:

1. The abstract is extremely detailed and dense. Are all the numbers necessary? A number of abbreviations are used, which should not be necessary in an abstract.

2. The overall hypothesis and conclusion of the paper seems to be that the Denver Cyclone contributes to aerosol concentrations in the Denver metro area primarily by

transporting aerosol and/or aerosol precursors from the nothern Front Range. This should be made much more clear in the abstract, introduction, and conclusions.

3. Relative humidity is used throughout the paper. I understand that RH is very important for aerosol properties, but it has a number of drawbacks as a meteorological variable. A conserved humidity variable such as specific humidity or mixing ratio is more appropriate. RH varies strongly with height in a well-mixed boundary layer because the temperature varies, while potential temperature and mixing ratio may be more or less constant with height. RH can only be compared at the same temperature. For example, check section 3.5.

4. The argument about relative increases in CO vs. ethane seems incorrect (section 3.3.1, last paragraph). CO has a small percentage increase because it has a large background, while ethane has no background. The correct comparison would be made by removing the CO background. The conclusion may not change. This propagates through to section 3.4, which should be checked for consistency.

Specific comments:

1. p.8, lines 27-30: The spatial contrast and separation are present but not "stark".

2. p.15, line 2: Again, "isolation" is too strong.

3. p.15, line 16: "Dramatically" is probably too strong. Can you make a quantitative estimate here?
* * *

---

## Referee Comment (RC2) · Anonymous Referee #2 · 2 Aug 2016

The authors have used part of the measurements during the FRAPPÉ study to evaluate the impacts of the Denver Cyclone on the local air quality based on meteorological variables, gaseous and aerosol measurements, some modeling, and comparison of results. This paper is well-written and outlines the details of the data analysis clearly to present the conclusion that the Denver Cyclone does indeed affect regional air pollution levels, especially in the Denver metro area. The data and analysis presented in this paper will be useful for future papers based on data collected during FRAPPÉ and other studies in this region.

Overall, the paper is good. I have some suggestions, outlined below, that make it more concise.

1. The abstract can be shortened without compromising the intended message. For example, the sentence "Average nitrate mass..., respectively." can be excluded from the abstract. Also, the way the abstract is written is just informative of the main text but not of the conclusion or the importance of the paper. After deleting some of the unnecessary information, it would be nice to add a sentence that addresses the importance and/or conclusions of the paper.

2. Adding a small table with the measurement dates and specifications (e.g., location) would be very helpful.

3. Page 3, second paragraph: I would suggest excluding this paragraph or making it more concise.

4. Page 4, first paragraph: This is a really good section of the introduction, but it gets lost in the current structure of the introduction. Re-structuring or making the introduction more concise will help bring this paragraph more attention.

5. Page 6, lines 4-10: Why do the authors emphasize the calibration procedures for the AMS, when they are using data from other instruments too? I suggest moving this paragraph to the supplementary material if the authors wish to keep it.

6. Page 9: Were i-pentane and n-pentane measured and could the authors use the ratio (or i/n butane) to discuss the O&G influence further?

7. Figure 4: If possible to do without cluttering the figure too much, it would be helpful to have an outline of the O&G rich area on one of the maps in this figure.

---

## Author Response (AR1)

We appreciate reviewer's time and effort for providing us with comments and suggestions on our manuscript. We have made the necessary revisions to the manuscript. Below, you will find our response and the summary of our approach, highlighted in *blue,* with modifications to the manuscript highlighted *in bold*:

*Referee #1*

The discussion paper describes measurements of gas phase and aerosol species over Colorado, and relates them to the Denver Cyclone. In general the measurements and analysis are good and well-described. The clarity of the presentation could be improved in several ways before final publication.

**General comments:**
1.   **The abstract is extremely detailed and dense. Are all the numbers necessary? A number of abbreviations are used, which should not be necessary in an abstract.**

     *The abstract has been revised to reflect the reviewer's suggestions:*
     ***"We present airborne measurements made during the 2014 Front Range Air Pollution and Photochemistry Éxperiment (FRAPPÉ) project to investigate the impacts of the Denver Cyclone on regional air quality in the greater Denver area. Data on trace gases, non-refractory sub-micron aerosol chemical constituents, and aerosol optical extinction ($\beta_{ext}$) at $\lambda = 632\,nm$ were evaluated in the presence and absence of the surface mesoscale circulation in three distinct study regions of the Front Range: In-Flow, Northern Front Range, and the Denver Metropolitan. Pronounced increases in mass concentrations of organics, nitrate, and sulfate in Northern Front Range and the Denver Metropolitan were observed during the cyclone episodes (27–28 July) compared to the non-cyclonic days (26 July, 02–03 August). Organic aerosols dominated the mass concentrations on all evaluated days, with a 45 % increase in organics on cyclone days across all three regions while the increase during the cyclone episode was up to ~ 80 % over the Denver Metropolitan. In the most aged air masses ($NO_x/NO_y < 0.5$), background organic aerosols over the Denver Metropolitan increased by a factor of ~ 2.5 due to transport from Northern Front Range. Furthermore, enhanced partitioning of nitric acid to the aerosol phase was observed during the cyclone episodes, mainly due to increased abundance of gas phase ammonia. During the non-cyclone events, $\beta_{ext}$ displayed strong correlations ($r = 0.71$) with organic and nitrate in the Northern Front Range and only with organics ($r = 0.70$) in the Denver Metropolitan, while correlation of $\beta_{ext}$ during the cyclone was strongest ($r = 0.86$) with nitrate over Denver. Mass extinction efficiency (MEE) values in Denver Metropolitan were similar under cyclone and non-cyclone days despite the dominant influence of different aerosol species on $\beta_{ext}$. Our analysis showed that the meteorological patterns associated with the Denver Cyclone increased aerosol mass loadings in the Denver Metropolitan area mainly by transporting aerosols and/or aerosol precursors from the northern regions, leading to impaired visibility and air quality deterioration."***

2.   **The overall hypothesis and conclusion of the paper seems to be that the Denver Cyclone aerosol and/or aerosol precursors from the northern Front Range. This should be made much clearer in the abstract, introduction, and conclusions.**

*We have integrated the overall hypothesis in the abstract, introduction, and conclusions of this manuscript. The following sentences have been added:*

*Abstract:* ***"Our analysis showed that the meteorological patterns associated with the Denver Cyclone increased aerosol mass loadings in the Denver Metropolitan area mainly by transporting aerosols and/or aerosol precursors from the northern regions, leading to impaired visibility and air quality deterioration."***

*Introduction* ***"More importantly, limited studies have evaluated the summertime air quality implications of the Denver Cyclone that results in transport of pollutants from the Northern Front Range to the urban center."***

*Conclusions:* ***"The meteorological conditions during a Denver Cyclone promote transport of aerosol constituents and their precursors from the northern Front Range into the Denver Metropolitan area, increasing aerosol mass loadings and reducing visibility****."*

3. **Relative humidity is used throughout the paper. I understand that RH is very important for aerosol properties, but it has a number of drawbacks as a meteorological variable. A conserved humidity variable such as specific humidity or mixing ratio is more appropriate. RH varies strongly with height in a well-mixed boundary layer because the temperature varies, while potential temperature and mixing ratio may be more or less constant with height. RH can only be compared at the same temperature. For example, check section 3.5.**

*Relative humidity is an important variable to consider for aerosols. Since flight segments were less than 2500 m in a well-mixed boundary layer, we believe it is appropriate to use surface RH as representative values for RH during the flight segments in the BL. However, we agree that the use of surface RH as a variable does not fully explain the meteorological changes in water content with changes in altitude. We have taken the recommendation of the referee to use a more conserved meteorological variable and have included a discussion on specific humidity in Section 3.1, along with associated plots referenced in the supplementary materials (Fig. S2-3) to understand the meteorological differences during the cyclone. The following has been added to Section 3.1:* ***"As shown in Fig. S3b-d air masses with higher water content were advected westward by easterly winds, ahead of the intensifying low pressure system that was developed by 18:00 UTC (11:00 MST on 27 July)."***
*Since RH is an important variable for aerosol partitioning and equilibrium, we kept Table 1 and the discussions related to ISORROPIA modeling in its original presentation.*

4. **The argument about relative increases in CO vs. ethane seems incorrect (section 3.3.1, last paragraph). CO has a small percentage increase because it has a large background, while ethane has no background. The correct comparison would be made by removing the CO background. The conclusion may not change. This propagates through to section 3.4, which should be checked for consistency.**

*The discussion on relative changes in carbon monoxide and ethane has been omitted from the manuscript. We believe a stronger supporting evidence for transport of O&G-influenced air masses south to the urban corridor is the observed change in the ratio of i-pentane to n-pentane*

*over DM during the cyclone (added now as Fig.7e). The last paragraph in section 3.3.1 has been replaced with the following:*

*"Mean mixing ratios of CO over DM during the cyclone were 144±23 ppbv compared to 110±8.7 ppbv in In-Flow and 114±12 ppbv in NFR. Additionally, mean values of CO and $C_2H_6$ in DM increased during the cyclone events compared to non-cyclone days (Fig. 7b,d). Since vehicular sources of CO are concentrated in DM, the slight increase in CO over DM during the cyclone was likely due to changes in the background CO in the region and a shallower morning boundary layer on 27-28 July. However, the increase in $C_2H_6$ could be due to release of emissions into a shallower morning boundary layer on cyclone days, the cyclonic mixing of air masses from northern latitudes with higher emissions of $C_2H_6$ from O&G operations, or a combination of these two phenomena. The observed increase in the mean $C_2H_6$ mixing ratio in DM during the cyclone compared to the non-cyclone days were 10.2±6.2 ppbv vs. 6.0±7.8 ppbv, respectively.  To better understand the influence of O&G operations over DM during the cyclone, we examined the ratio of i-pentane to n-pentane since O&G emissions show a characteristic ratio in the range of 0.8 – 1.2 (Gilman et al., 2013; Swarthout et al., 2013; Thompson et al., 2014; Halliday et al., 2016) in contrast to urban sources predominately impacted by vehicular emissions, which typically have a higher ratio between 2-3 (Broderick and Marnane, 2002; Baker et al., 2008).  Figure 7e represents the statistical analysis of i-pentane to n-pentane ratio in the threes study regions. Non-cyclone days show a significant urban source of pentanes in DM compared to NFR. During the cyclone, a minor decrease in the ratio was observed in NFR, whereas the ratio decreased substantially in DM to values close to those in NFR. These observations suggest that the significant increase in $C_2H_6$ mixing ratio observed over DM during the cyclone cannot be solely explained by BL height differences, but rather driven by transport of O&G-impacted and $C_2H_6$-rich air masses from NFR into the DM."*

**Specific comments:**

1.  **p.8, lines 27-30: The spatial contrast and separation are present but not "stark".**

    *Then sentence has been modified as following: "Consistent with the meteorological conditions presented above, there is a contrast in the spatial distribution and separation of pollutants during the non-cyclone and cyclone situations."*

2.  **p.15, line 2: Again, "isolation" is too strong.**

    *Then sentence has been modified as following:  "Overall, particle formation and growth during the non-cyclonic episodes occurred predominantly downwind of the major point/area sources"*

3.  **p.15, line 16: "Dramatically" is probably too strong. Can you make a quantitative estimate here?**

    *Then sentence has been modified as following: "Based on these results, reduction in source strengths of aerosol precursors in NFR leading to OA and ammonium nitrate formation, including mitigation of $NH_3$ emissions from dairy and livestock farming, could  effectively*

*reduce the impact of cyclone events on Denver's air quality by reducing the aerosol mass loadings by a factor of 2 (i.e., ~11 μg sm$^{-3}$ to 5 μg sm$^{-3}$) and improving visibility by approximately 3 folds (i.e., ~32 Mm$^{-1}$ to 11 Mm$^{-1}$)."*

We appreciate reviewer's time and effort for providing us with comments and suggestions on our manuscript. We have made the necessary revisions to the manuscript. Below, you will find our response and the summary of our approach, highlighted in *blue,* with modifications to the manuscript highlighted *in bold*:

*Referee#2*

The authors have used part of the measurements during the FRAPPÉ study to evaluate the impacts of the Denver Cyclone on the local air quality based on meteorological variables, gaseous and aerosol measurements, some modeling, and comparison of results. This paper is well-written and outlines the details of the data analysis clearly to present the conclusion that the Denver Cyclone does indeed affect regional air pollution levels, especially in the Denver metro area. The data and analysis presented in this paper will be useful for future papers based on data collected during FRAPPÉ and other studies in this region. Overall, the paper is good. I have some suggestions, outlined below, that make it more concise.

1. **The abstract can be shortened without compromising the intended message. For example, the sentence "Average nitrate mass. . ., respectively." can be excluded from the abstract. Also, the way the abstract is written is just informative of the main text but not of the conclusion or the importance of the paper. After deleting some of the unnecessary information, it would be nice to add a sentence that addresses the importance and/or conclusions of the paper.**

   *The abstract has been revised to reflect the reviewer's suggestions:*
   ***"We present airborne measurements made during the 2014 Front Range Air Pollution and Photochemistry Éxperiment (FRAPPÉ) project to investigate the impacts of the Denver Cyclone on regional air quality in the greater Denver area. Data on trace gases, non-refractory sub-micron aerosol chemical constituents, and aerosol optical extinction ($\beta_{ext}$) at $\lambda = 632\,nm$ were evaluated in the presence and absence of the surface mesoscale circulation in three distinct study regions of the Front Range: In-Flow, Northern Front Range, and the Denver Metropolitan. Pronounced increases in mass concentrations of organics, nitrate, and sulfate in Northern Front Range and the Denver Metropolitan were observed during the cyclone episodes (27–28 July) compared to the non-cyclonic days (26 July, 02–03 August). Organics aerosols dominated the mass concentrations on all evaluated days, with a 45 % increase in organics on cyclone days across all three regions while the increase during the cyclone episode was up to ~ 80 % over the Denver Metropolitan. In the most aged air masses ($NO_x/NO_y < 0.5$), background organic aerosols over the Denver Metropolitan increased by a factor of ~ 2.5 due to transport from Northern Front Range. Furthermore, enhanced partitioning of nitric acid to the aerosol phase was observed during the cyclone episodes, mainly due to increased abundance of gas phase ammonia. During the non-cyclone events, $\beta_{ext}$ displayed strong correlations (r = 0.71) with organic and nitrate in the Northern Front Range and only with organics (r = 0.70) in the Denver Metropolitan, while correlation of $\beta_{ext}$ during the cyclone was strongest (r = 0.86) with nitrate over Denver. Mass extinction efficiency (MEE) values in Denver Metropolitan were similar under cyclone and non-cyclone days despite the dominant influence of different aerosol species on $\beta_{ext}$. Our analysis showed that the meteorological***

*patterns associated with the Denver Cyclone increased aerosol mass loadings in the Denver Metropolitan area mainly by transporting aerosols and/or aerosol precursors from the northern regions, leading to impaired visibility and air quality deterioration."*

2. **Adding a small table with the measurement dates and specifications (e.g., location) would be very helpful.**

An additional flight map has been added to the supplemental materials section (Fig. S1) that depicts flight tracks of the entire field campaign with their corresponding, flight number, dates, and locations of the active O&G wells.

[Figure]

3. **Page 3, second paragraph: I would suggest excluding this paragraph or making it more concise.**

Our goal to include such a paragraph in the Introduction was to highlight the major studies previously carried out in the region and to compare our results and put the current summertime measurements in context. This paragraph has now been revised and shortened, as following:

*"Emission sources and meteorological conditions affecting air quality in the greater Front Range have been previously studied in the region. The 1973 Denver Air Pollution Study (Russell, 1976), focused on episodes of winter pollution in Denver, described occurrences of rapid dispersal of pollutants to the north-northeast of Denver due to strong winds and recurring reversal of winds, bringing aged pollutants back to the urban center. Additionally, the Denver Haze Study conducted in the winter of 1978-1979 and the 1987-88 Metro Denver Brown Cloud study provided objective apportionment to the observed brown cloud pollution*

*over Denver. The occurrence of the wintertime inversion layer and emissions from the local gas and coal burning power plants had a profound effect on air quality and visibility degradation. Among the measured aerosol species, elemental carbon, ammonium sulfate, and ammonium nitrate contributed to the majority of optical extinction, decreasing visibility in the visible range by about 38%, 20%, and 17%, respectively (Countess et al., 1980; Groblicki et al., 1981; Wolff et al., 1981; Watson et al., 1988; Neff, 1989)."*

4. **Page 4, first paragraph: This is a really good section of the introduction, but it gets lost in the current structure of the introduction. Re-structuring or making the introduction more concise will help bring this paragraph more attention.**

   The referenced paragraph has been moved to an earlier section in the Introduction to explain the meteorology in the Front Range before discussing previous measurements.

5. **Page 6, lines 4-10: Why do the authors emphasize the calibration procedures for the AMS, when they are using data from other instruments too? I suggest moving this paragraph to the supplementary material if the authors wish to keep it.**

   Since observed aerosol concentrations are a major focus of this manuscript and because of the recent discussions about the quantification limits of the AMS instrument, we choose to keep this information in the main text for completeness and to provide the AMS users with the necessary operational and sensitivity related details. We have re-structured the above referenced lines to be included in section 2.2, following the introduction of the AMS instrument.

6. **Page 9: Were i-pentane and n-pentane measured and could the authors use the ratio (or i/n butane) to discuss the O&G influence further?**

   *This is a great suggestion. We examined the ratio of i-pentane to n-pentane and indeed very different ratios were observed in DM during the cyclone and non-cyclone days due to the influence of O&G emissions. We have added a discussion on this in Section 3.3.1 and statistics of the ratio in Fig. 7e.*

   ***"To better understand the influence of O&G operations over DM during the cyclone, we examined the ratio of i-pentane to n-pentane since O&G emissions show a characteristic ratio in the range of 0.8 – 1.2 (Gilman et al., 2013; Swarthout et al., 2013; Thompson et al., 2014; Halliday et al., 2016) in contrast to urban sources predominately impacted by vehicular emissions, which typically have a higher ratio between 2-3 (Broderick and Marnane, 2002; Baker et al., 2008). Figure 7e represents the statistical analysis of i-pentane to n-pentane ratio in the threes study regions. Non-cyclone days show a significant urban source of pentanes in DM compared to NFR. During the cyclone, a minor decrease in the ratio was observed in NFR, whereas the ratio decreased substantially in DM to values close to those in NFR. These observations suggest that the significant increase in $C_2H_6$ mixing ratio observed over DM during the cyclone cannot be solely explained by BL height differences, but rather driven by transport of O&G-impacted and $C_2H_6$-rich air masses from NFR into the DM."***

**(e)**

[Figure]

7. **Figure 4: If possible to do without cluttering the figure too much, it would be helpful to have an outline of the O&G rich area on one of the maps in this figure.**

   *Figure 1, 4-6a, and S1 now include markers that represent active oil and gas wells in the Colorado Front Range for reference.*

**Impacts of the Denver Cyclone on Regional Air Quality and Aerosol Formation in the Colorado Front Range during FRAPPÉ 2014**

Kennedy T. Vu[1], Justin H. Dingle[1], Roya Bahreini[1,2], Patrick J. Reddy[3†], Eric C. Apel[3], Teresa L. Campos[3], Joshua P. DiGangi[4], Glenn S. Diskin[4], Alan Fried[5], Scott C. Herndon[6], Alan J. Hills[3], Rebecca S. Hornbrook[3], Greg Huey[7], Lisa Kaser[3], Denise D. Montzka[3], John B. Nowak[6], Sally E. Pusede[8], Dirk Richter[5], Joseph R. Roscioli[6], Glen W. Sachse[9], Stephen Shertz[3], Meghan Stell[3], David Tanner[7], Geoffrey S. Tyndall[3], James Walega[5], Peter Weibring[5], Andrew J. Weinheimer[3], Gabriele Pfister[3], Frank Flocke[3]

[1] Environmental Toxicology Graduate Program, University of California, Riverside, CA 92521
[2] Department of Environmental Sciences, University of California, Riverside, CA 92521
[3] Atmospheric Chemistry Observations & Modeling Laboratory, National Center for Atmospheric Research, Boulder, CO 80301
[4] Chemistry and Dynamics Branch, NASA Langley Research Center, Hampton, VA 23681
[5] Institute for Arctic and Alpine Research, University of Colorado, Boulder, CO 80303
[6] Aerodyne Research, Inc., Billerica, MA 01821
[7] Department of Earth and Atmospheric Sciences, Georgia Institute of Technology, Atlanta, GA 30033
[8] Department of Environmental Sciences, University of Virginia, Charlottesville, VA 22904
[9] National Institute of Aerospace, Hampton, VA 23666
[†] Visitor at NCAR, Boulder, CO 80301

*Correspondence to*: R. Bahreini (Roya.Bahreini@ucr.edu)

**Abstract.** We present airborne measurements made during the 2014 Front Range Air Pollution and Photochemistry Éxperiment (FRAPPÉ) project to investigate the impacts of the Denver Cyclone on regional air quality in the greater Denver area. Data on trace gases, non-refractory sub-micron aerosol chemical constituents, and aerosol optical extinction ($\beta_{ext}$) at $\lambda = 632$ nm were evaluated in the presence and absence of the surface mesoscale circulation in three distinct study regions of the Front Range: In-Flow, Northern Front Range, and the Denver Metropolitan. Pronounced increases in mass concentrations of organics, nitrate, and sulfate in Northern Front Range and the Denver Metropolitan were observed during the cyclone episodes (27–28 July) compared to the non-cyclonic days (26 July, 02–03 August). Organic aerosols dominated the mass concentrations on all evaluated days, with a 45 % increase in organics on cyclone days across all three regions while the increase during the cyclone episode was up to ~ 80 % over the Denver Metropolitan. In the most aged air masses ($NO_x/NO_y < 0.5$), background organic aerosols over the Denver Metropolitan increased by a factor of ~ 2.5 due to transport from Northern Front Range. Furthermore, enhanced partitioning of nitric acid to the aerosol phase was observed during the cyclone episodes, mainly due to increased abundance of gas phase ammonia. During the non-cyclone events, $\beta_{ext}$ displayed strong correlations ($r = 0.71$) with organic and nitrate in the Northern Front Range and only with organics ($r = 0.70$) in the Denver Metropolitan, while correlation of $\beta_{ext}$ during the cyclone was strongest ($r = 0.86$) with nitrate over Denver. Mass

extinction efficiency  (MEE) values in Denver Metropolitan were similar under cyclone and non-cyclone days despite the dominant influence of different aerosol species on $\beta_{ext}$. Our analysis showed that the  meteorological patterns associated with the Denver Cyclone increased aerosol mass loadings in the Denver Metropolitan area mainly by transporting aerosols and/or aerosol precursors from the northern regions, leading to impaired visibility and air quality deterioration.

~~the Data on trace gases, non-refractory sub-micron aerosol chemical constituents, and aerosol optical extinction ($\beta_{ext}$) at $\lambda$=632 nm in the presence and absence of a surface mesoscale circulation pattern, called the Denver Cyclone, were analyzed in three study regions of the Front Range: In-Flow, Northern Front Range, and Denver Metropolitan. Pronounced increases in mass concentrations of organics, nitrate, and sulfate in NFR and DM were observed during the cyclone episodes (27-28 July) compared to the non-cyclonic days (26 July, 02-03 August). Organic aerosols dominated the mass concentrations on all evaluated days, with a 45% increase in on cyclone days across all three regions. In the most aged air masses ($NO_x/NO_y$ <0.5), background organic aerosols over the Denver Metropolitan increased by a factor of ~4, from 0.93±0.33 μg sm$^{-3}$ to 3.70±0.28 μg sm$^{-3}$ due to transport from NFR. Furthermore, enhanced partitioning of nitric acid to the aerosol phase was observed during the cyclone episodes, mainly due to increased abundance of gas phase $NH_3$. During the non-cyclone events, extinction displayed strong correlations (r=0.71) with organic aerosols and nitrate in the Northern Front Range while correlation of extinction 
[revised manuscript text omitted]
.*, 2008).~~non-methane hydrocarbon ratios as indicators to differentiate between air mass types. Iso pentane and n pentane are co emitted from different sources, but pentanes have similar reaction rate constants so the mixing ratios of iso pentane to n pentane are preserved under atmospheric aging. Several pentane source emissions have been studied (Baker *et al.*, 2008; Simpson *et al.*, 2010; Simpson *et al.*, 2011)(Pétron *et al.*, 2012; Gilman *et al.*, 2013) in contrast to urban sources which typically have a hi7dwith respect tothewith respectby nearly 505%pentane emissionsthe,a~50% decreasesuggests a present O&G influence over the region.~~ 
[revised manuscript text omitted]